# Spice and Herb Frauds: Types, Incidence, and Detection: The State of the Art

**DOI:** 10.3390/foods12183373

**Published:** 2023-09-08

**Authors:** Rocío Velázquez, Alicia Rodríguez, Alejandro Hernández, Rocío Casquete, María J. Benito, Alberto Martín

**Affiliations:** 1Departamento de Ingeniería, Medio Agronómico y Forestal, Investigación Aplicada en Hortofruticultura y Jardinería, Escuela de Ingenierías Agrarias, Universidad de Extremadura, Avda. Adolfo Suárez s/n, 06007 Badajoz, Spain; rvotero@unex.es; 2Instituto Universitario de Investigación de Recursos Agrarios (INURA), Universidad de Extremadura, Avda. de la Investigación s/n, Campus Universitario, 06006 Badajoz, Spain; ahernandez@unex.es (A.H.); rociocp@unex.es (R.C.); mjbenito@unex.es (M.J.B.); amartin@unex.es (A.M.); 3Departamento de Producción Animal y Ciencia de los Alimentos, Nutrición y Bromatología, Escuela de Ingenierías Agrarias, Universidad de Extremadura, Avda. Adolfo Suárez s/n, 06007 Badajoz, Spain

**Keywords:** spices, herbs, fraud, adulteration, adulterants, detection methods, quality

## Abstract

There is a necessity to protect the quality and authenticity of herbs and spices because of the increase in the fraud and adulteration incidence during the last 30 years. There are several aspects that make herbs and spices quite vulnerable to fraud and adulteration, including their positive and desirable sensorial and health-related properties, the form in which they are sold, which is mostly powdered, and their economic relevance around the world, even in developing countries. For these reasons, sensitive, rapid, and reliable techniques are needed to verify the authenticity of these agri-food products and implement effective adulteration prevention measures. This review highlights why spices and herbs are highly valued ingredients, their economic importance, and the official quality schemes to protect their quality and authenticity. In addition to this, the type of frauds that can take place with spices and herbs have been disclosed, and the fraud incidence and an overview of scientific articles related to fraud and adulteration based on the Rapid Alert System Feed and Food (RASFF) and the Web of Science databases, respectively, during the last 30 years, is carried out here. Next, the methods used to detect adulterants in spices and herbs are reviewed, with DNA-based techniques and mainly spectroscopy and image analysis methods being the most recommended. Finally, the available adulteration prevention measurements for spices and herbs are presented, and future perspectives are also discussed.

## 1. Introduction

### 1.1. Herbs and Spices Sensorial and Health-Related Properties

Herbs and spices constitute a large group of plant materials whose terms refer to different parts of the plants; herbs are related to the green parts of the plant, while spices are applicable to other parts such as bulbs, roots, bark, flowers, seeds, etc. There is a great diversity of herbs and spices, an example of which is the List of Culinary Herbs and Spices, provided by The European Spices Association [1], which includes frequently traded dried herbs and spices: 33 herbs with use of leaves and 53 spices with use of fruits and seeds.

Herbs and spices are highly valued as ingredients for their aromatic, flavouring and colouring properties, in addition to their functional properties, since they contain a wide range of active molecules belonging to polyphenols, flavonoids, terpenoids, and sulphur compounds, among others. Among functional activities, it is possible to outline the intense antioxidant activity of spices such as turmeric (*Curcuma longa* L.), ginger (*Zingiber officinale* Roscoe), lemon grass (*Cymbopogon citratus* (hort. ex DC.) Stapf) and clove (*Syzygium aromaticum* L. Merr. & L.M. Perry), mainly associated with total phenolic compounds and flavonoids [2,3]. Other herbs and spices have interesting anti-inflammatory properties, such as turmeric and ginger [4]; herbal teas and cinnamon [5]; chili pepper *(Capsicum annuum* L.), allspice (*Pimenta dioica* L.), basil (*Ocimum basilicum* L.), bay leaves (*Laurus nobilis* L.), black pepper (*Piper nigrum* L.), liquorice (*Glycyrrhiza glabra* L.), nutmeg (*Myristica fragrans* Houtt.), oregano (*Origanum vulgare* L.), sage (*Salvia officinalis* L.) and thyme (*Thymus vulgaris* L.) [6]. In addition, herbs and spices have been extensively studied for the prevention and treatment of multiple diseases. In this regard, the anticarcinogenic capacities of spices such as ginger and black cumin (*Nigella sativa* L.) [7] are particularly noteworthy, although many herbs and spices have been described as having activity against various types of cancer [8]. Due to their properties for the treatment of various diseases, herbs and spices have been used since ancient times in traditional medical systems; for example, ayurvedic medicine has been extensively reviewed by numerous authors (e.g., it was recently reviewed by [9,10,11,12,13]). On the other hand, another remarkable activity of a large number of herbs and spices is the ability to control microbial growth, avoiding undesirable contamination of food and feed. In this regard, a high capacity to control pathogenic Gram-positive and Gram-negative bacteria such as, for example, *Listeria monocytogenes*, *Escherichia coli*, *Bacillus cereus* and *Staphylococcus aureus* has been demonstrated using extracts of oregano, clove, sage, rosemary, etc. [14], while their antifungal activities are more limited [15]. All these positive and desirable properties that herbs and spices have and the form in which they are sold, mostly powered, make these ingredients so vulnerable to fraud and adulteration.

### 1.2. World Market of Herbs and Spices: Economic Relevance 

At the socio-economic level, herbs and spices are of great relevance, especially in Asia and Africa. The Food and Agriculture Organization (FAO) of the United Nations provides accurate information on a small group of herbs and spices, those with greater relevance from the productive and economic points of view, which are shown in Table 1 [16,17]: anise, badian, coriander, cumin, caraway, fennel and juniper berries; dried chillies and peppers; cinnamon and cinnamon-tree flowers; cloves; ginger; mustard; nutmeg, mace, cardamoms; other stimulant, spicy and aromatic crops and peppers; peppermint and spearmint; and sesame seeds.

Regarding the group of herbs, peppermint and spearmint stood out, with almost 0.4 million t in the world, with the main producing country being Morocco (78.9%), followed by Argentina (18.3%). With respect to spices, the most important worldwide were sesame (6.35 Mill t), mainly produced in Sudan (17.6%), India (12.9%) and Tanzania (11.0%); ginger (4.90 Mill t) in India (45.4%), Nigeria (15.7%) and China (13.0%); and chillies and peppers (4.84 Mill t) in India (42.3%), Bangladesh (10.2%) and Thailand (6.9%) [16].

In relation to trade data, in 2021 imports of herbs and spices were USD 17,684.29 million, highlighting sesame (20.2%), dried chilies and peppers (15.7%), other stimulant, spicy and aromatic crops (12.6%), pepper (11.5%) and ginger (8.5%), among others [17].

Globally, in 2021, 21.14 million ha of herbs and spices were cultivated, with a total production of 23.87 million tons. The main producing continent of herbs and especially spices was Asia, with 68.8% of world production, followed by Africa with 26.2%, while production was lower in America (3.8%), Europe (1.2%) and Oceania (0.1%) [16]. It should be noted that Asian countries are the largest producers of spices, with India accounting for 36% of world production [16]. 

The production volumes of exports and imports, at the global level, were 6.97 and 7.22 million t (Table 2). Likewise, the economic values of exports and imports amounted to 17,012.65 and 17,684.30 USD Mill., respectively. Asia is the largest producer and consumer of herbs and spices, as well as having a high volume of trade, with both exports and imports exceeding 4 million t and USD 9300 Mill. Africa, being the second largest producer of herbs and spices, also consumes a large part of what it produces and exports 1.4 million t, exceeding USD 2600 Mill. America has productions close to 1 million t, with exports and imports of > 0.7 million t and > USD 2000 million and 0.9 million t and > USD 3000 million, respectively. Europe only produces 0.3 million t of herbs and spices; exports are important (> 0.7 million t and > USD 2500 Mill.), but imports are more prevailing (>1.2 million t and > USD 4300 Mill.) [17].

### 1.3. Quality Protection of Herbs and Spices

Herbs and spices are closely linked to the geographical areas in which they are grown, with differences in quality and price depending on where they come from. Therefore, the physical–chemical characteristics and quality of these products rely on numerous factors, such as the plant material itself, the physical environment (climate, soil, and water), cultivation techniques (irrigation, fertilization, etc.) and the processing and/or transformation processes, among others. It is therefore of great importance to verify the origin and authenticity of those herbs and spices produced in a localized geographical area using traditional production and processing methods, which result in a product of differentiated quality.

Currently, in the European Union, there is a great diversity of agri-food products of recognized quality at market and consumer level. These products with a differentiated and high quality, mainly due to their origin, can be subject to usurpation and imitation. Quality schemes such as PDO (Protected Designation of Origin) and PGI (Protected Geographical Indication) [18], within the European Union, make it possible to protect agricultural and food products against any misuse, imitation, or evocation, and against any other practice that could mislead the consumer as to the true origin of the product. 

Among these products of differentiated quality, 19 are spices currently registered under Geographical Indications [19]: 13 are registered under the PDO European Union, 2 under the PGI European Union and 4 under the PGI non-EU countries (Table 3). Most of the PDO European Union are spices (mainly saffron and paprika) whose quality and characteristics are linked to geographical environments, while in other third countries, mainly Asian countries, other spices such as pepper, ginger and cinnamon are those registered under these quality schemes (Table 3).

## 2. Types of Frauds

In this context, where herbs and spices are highly valued for their culinary and functional properties, as well as having a high economic value often supported by recognised quality qualifications, fraudulent practices to increase trade margins are a serious issue for the sector. Herbs and spices are highly vulnerable to food fraud, due to their high level of global trade, and are products that require increased regulation and vigilance. Several organisations around the world have implemented regulatory standards to control the quality and authenticity of herbs and spices and to detect possible adulterants. The most prominent are the American Spice Trade Association (ASTA), the European Spice Association (ESA), the International Organisation of Spice Trade Association (IOSTA), the Spices Board of India, and the All Nippon Spice Trade Association (ANSA). In the framework of the EU legislation on official controls in food and feed (Regulation EU 2017/625) [20], ‘fraud notifications’ reported in the Rapid Alert System for Food and Feed (RASFF) of the European Commission means “a non-compliance concerning any suspected intentional action by businesses or individuals, for the purpose of deceiving purchasers and gaining undue advantage therefrom”. The rules in force in EU member states for suspected fraud to be reported as “non-compliance” or “suspicion of fraud” include: (1) Violation of EU rules: a violation of one or more rules laid out in the EU agri-food chain legislation, as referred to in Article 1(2) of Regulation (EU) 2017/625 [20]; (2) Deception of customers: Some form of deception of the customers/consumers (for example: altered colouring or altered labels, which hide the true quality or, in worse cases, even the nature of a product). Moreover, the deceptive element may also come as a public health risk, as some of the real properties of the product are hidden (for example, in the case of undeclared allergens); (3) Undue advantage: the fraudulent act brings some form of direct or indirect economic advantage for the perpetrator; and (4) Intention: verified when a number of factors give strong grounds to show that certain non-compliances are not accidental, such as the intentional substitution of a high-quality ingredient with a lower-quality one, rather than an accidental contamination due to the production process. 

Technical Report Implementation of Article 9(2) of Regulation (EU) 2017/625 [20,21] defines different types of fraud:Adulteration and product tampering: Addition of a foreign or inferior-quality substance or element; by replacing a more valuable substance or element with less valuable or inert ingredients, so that they no longer match the implicit or explicit claims associated with the agri-food product. Adulterations can be carried out by the following actions: substitution, dilution, removal, unapproved/undeclared enhancement and concealment, and unapproved/undeclared treatment, process, or product. In the case of the addition of components, these could reduce the quality and alter the composition of the food itself, potentially causing health risks to consumers [22].Counterfeit: Intellectual Property Rights (IPR) infringement, including any aspects of the genuine agri-food product or packaging being replicated, for instance, the process of copying the brand name, packaging concept, or processing method for economic gain.Document forgery: The process of creating, adapting, altering, misrepresenting, or imitating documents such as certificates, passports, analytical test reports, declarations of compliance, and other identification, and administrative documents.Grey market activities: Production, theft, and diversion involving unauthorized sales channels for agri-food products (traceability issues).Misdescription/mislabelling/misbranding: Placing of explicit false claims or distorting the information on the label/packaging of expiry/production date, nutrition/health claims, geographical claims (excluding PGO, PDI, TSG), quality terms, and/or quantity (weight and volume).

In the herbs and spices sector, adulteration by incorporation of non-declared or non-permitted components is of great relevance. It consists of the inclusion of any substance not legally declared, not authorized, or present in a manner likely to mislead the consumer, being an imitated and/or reduced quality product. Fraudulent adulteration practices [23] include:A different part of the same botanical plant, rather than the one declared, to the extent that this would mislead the customer.

In this type of intentional adulteration, the plant part of interest is replaced by other parts of inferior quality, with fewer bioactive compounds and which is therefore cheaper, thus achieving greater economic benefit. Among other common herbs and spices, the addition of sage and laurel stems has been detected instead of clove leaves and stems, which contain a lower percentage of essential oils than the flower buds of this spice [24]. In addition, parts of the same plant can be added to increase the weight or volume of spices [25,26], for example, the addition of non-spice plant matter such as stamens and safflower to pure saffron [27,28]. This type of adulteration practice is one of the most difficult to detect. It is of utmost importance to know the profile and concentration of bioactive compounds of a spice or herb and to establish thresholds for such a specific spice or herb, discarding those that do not meet these criteria.

Technically avoidable amounts of parts from other botanical plants than the one declared.

This is another deliberate fraudulent practice, in which the plant material of interest is replaced by a different plant species, usually cheaper than the herb or spice to be marketed. In addition, some of these fraudulent manipulations also add plant species of inferior quality, with poorer organoleptic properties and lower content of active ingredients, and some of them can even cause toxicity in consumers. There are some examples, in the case of black pepper (*P. nigrum* L.), one of the most widely used spices in the world, which can be adulterated with papaya seeds (*Carica papaya* L.), which have very similar external characteristics [29], and which can cause liver and stomach problems [30]. Another representative example of this type of fraud is genuine cinnamon (*Cinnamomum verum* J. Presl) which is adulterated with inferior-quality cinnamon commonly known as cassia (*Cinnamomum cassia* (L.) J. Presl), which is less aromatic and has a high coumarin content that can lead to toxicity problems, and is regarded as a possible genotoxic carcinogen [31]. 

Ingredients, additives, dyes, or any other constituent not approved for use in herbs and spices.

In this case, filler adulterants of an organic nature may be used, notably the addition of foreign matter with similar physical characteristics (colour and particle size), and with a lower economic value than the spice of interest. For example, the use of cereal and potato starch is included in some spice powders such as paprika, curry, turmeric, and ginger. In addition, corn starch is added as an adulterant in onion powder [32,33], garlic and ginger powders [34]. Other common adulterants used in ground cinnamon are different bulking agents such as flour and coffee husks [35]. Also, the addition of plant fillers with similar morphological characteristics to the herbs of interest has also been detected, such as the presence of olive leaves (*Olea europaea* L.) in some ground herbs such as oregano and sage [36]. Oregano is frequently adulterated with olive or myrtle leaves [37], including sumac, cistus, and hazelnut leaves. Other adulteration practices in herbs and spices include artificial chemicals, in spices pre-diluted with vegetable fillers, mainly to enhance the flavour of the spices. This type of adulteration is common in cinnamon bark powder adulterated with pepper powder, cinnamon oil, clove powder, clove oil and a commercial preparation containing cinnamaldehyde and eugenol [38]. Piperine has been detected in black pepper and in other spice compositions [39]. In addition, in authentic vanilla (*Vanilla planifolia* Jacks.), the use of fraudulent vanilla extract using synthetic vanillin that is cheaper than natural vanilla extract is common [40].

In other cases of adulteration, non-food materials are added, which are inorganic in nature, very economical and contain contaminants that negatively affect the health of consumers. For example, the use of brick dust to adulterate paprika powder [41], sawdust and stone dust in chilli powder [42] or yellow chalk powder in turmeric [43]. Other adulterations include the addition of lead oxide to paprika powder [44] and red lead oxide to cayenne pepper [45]. On the other hand, the use of dyes harmful to the health of consumers is also illegally added to spices. Sudan dyes are synthetic dyes used in the manufacture of plastics, textiles, etc., classified as carcinogenic in humans and animals and therefore banned from being added to food. The addition of unauthorized dyes in spices carries serious health risks, and this has been addressed extensively in the literature. The inclusion of Sudan dyes in powdered products such as red pepper, chilli, turmeric, paprika, and saffron has been detected [25,46]. Saffron can be adulterated by adding synthetic dye substances such as quinoline, sunset yellow, Sudan II, Allura red and tartrazine [47].

Ingredients, additives, dyes, or any other constituent approved for use in food but unlawfully not declared or indicated in a form which might mislead the customer.

In this type of adulteration, fraudulent ingredients may have a high allergenic potential. Most allergic reactions to herbs and spices are due to the fraudulent inclusion, either intentionally or accidentally (by cross-contamination) of milk, eggs, nuts (peanuts), and cereals (wheat) which may cause adverse health effects in those allergic to these products. The presence of undeclared allergens has been detected in different spices, such as peanut and almond protein in cumin [48], casein (milk protein), ovalbumin (egg protein), protein gluten (wheat flour) and peanut in curry powder [49] and gluten (wheat, rye, and barley cereal flours) in chilli curry powder [50]. Ground chilli spice can be adulterated with dried red beet pulp and powdered *Z. nummularia* (Burm. f.) Wight & Arn. fruits [51] or with groundnut or almond shell residues which can cause health-related issues for the consumer. For example, cases of allergy or anaphylaxis have been reported after consumption of cumin and paprika adulterated with nut protein [48,52].

In addition, there are fraudulent practices in which natural dyes from different plants are added: for example, a dye extracted from the flowers of *Buddleja officinalis* Maxim. used to adulterate saffron [28].

In relation to the use of additives, the presence of undeclared additives (colouring agents and sodium benzoate) in different spices (spice mix, sumac, spice, dried pepper, chili) from Asian and Middle Eastern countries has been detected [53].

Herbs and spices that have had any valuable constituent omitted or removed which misleads the customer (e.g., spent and partially spent herbs and spices, de-oiled material, and defatted material).

This type of adulteration consists of replacing, partially or completely, commercial herbs or spices with spent herbs or spices: for example, the addition of waste products such as dried tomato peel to adulterate paprika powder [54]. Adulteration is notable with the addition of spices that have had their valuable components removed, such as the inclusion of defatted paprika in paprika [55]. Paprika oil (oleoresin) is a quality product with multiple health benefits. However, once removed from paprika, the remaining spent product can be used to adulterate paprika. Once this oleoresin is removed from paprika, the remaining “spent” material is a waste product [56].

On the other hand, foodstuffs (including herbs and spices) are considered to be adulterated when the labelling is incorrect due to various illegal practices: addition of undeclared material (adulterations already indicated above, mostly by partial or total substitution of the herb or spice of interest by cheaper components); incorrect declaration of production methods (e.g., “organic production” is indicated for conventional production, e.g., organic vanilla with a fraudulent “organic” certification [29]; misrepresentation of the product’s production methods (e.g., “sun-dried” for “hot air dried”); inappropriate handling of expiry dates; and misrepresentation of the geographical origin of products [57].

The most adulterated spices in terms of origin are paprika (*Capsicum annuum* L.), black pepper (*P. nigrum* L.), cinnamon (*C. verum* J. Presl), turmeric (*C. longa* L.) and saffron (*Crocus sativus* L.), while the most adulterated herb is oregano (*O. vulgare* L.) [35].

## 3. Fraud Incidence

At the EU level, RASFF provides information on the detection of health risks in food products, including herbs and spices. To understand fraud and adulteration in these products, the health notifications provided by the RASFF system have been analysed for the historical series from 1989 to 2020 [58]. 

In this 33-year period, 61,200 health notifications were issued for food products. A total of 3112 notifications (5.1%) concerned herbs and spices, of which 798 notifications (25.6%) were related to fraud and adulteration. The percentages of notifications for each type of adulteration were as follows: novel food (1%), allergens (7%), composition (63%), food additives and flavourings (13%), labelling (2%) and adulteration-fraud (13%). These data highlight the high number of adulterations related to composition, i.e., the use of unauthorised dyes (Figure 1). Going more into detail, only 11 unauthorised novel food ingredients were reported, such as olive leaves in oregano, *Angelica sinensis* (Oliv.) in a spice mix, hemp flowers in hemp spices for herb butter, frozen seasoned perilla leaves (*Perilla frutescens* L.), stevia leaves and powdered stevia leaves, kava kava (*Piper methysticum* L.), or *Siraitia grosvenorii* (Swingle) in spices, among others. In the allergen group, almost 60 notifications were recorded in pure spices or spice mixtures with the presence of other undeclared plant products, mainly undeclared celery (26%), undeclared mustard (26%), traces of almonds (10%), traces of peanut (16%), undeclared wheat (5%), gluten (10%), and other undeclared products (7%).

However, the most important type of adulteration refers to the use of different unauthorised synthetic dyes, with more than 500 composition notifications, where several unauthorised dyes are frequently detected in the same spice. The percentage of notifications in spices for the different unauthorised dyes is as follows: Butter Yellow (1%), Fast Garnet GBC (1%), Orange II (3%), Para Red (10%), Rhodamine B (2%), Sudan (79%), other unauthorised colours (2%) and other compositions (3%). The use of unauthorised colour dyes is notable in spices that stand out for their colour and colour intensity, which is why these products lend themselves to fraud, as shown by the number of notifications in the historical study series: curry (56 notifications), chilli pepper (134 notifications), paprika (59 notifications), cayenne pepper (12 notifications) and spice mixtures (83 notifications). In addition, the most-used dyes in adulterations are Sudan dyes, mainly Sudan I and Sudan IV, which are present in 65% and 33%, respectively, of the notifications referring to this type of dye.

The presence of food additive and flavouring notifications in herbs and spices is also marked in those spices which are attractive because of their colour. More than 100 notifications were recorded, mainly in spices, which presented the following additives: colour E 100—Curcumin (2%), colour E 102—tartrazine (10%), colour E 110—Sunset Yellow FCF (5%), colour E 122—azorubine (3%), colour E 123—amaranth (1%), colour E 124—Ponceau 4R (14%), colour E 127—erythrosine (8%), colour E 129—Allura Red (5%), AC, colour E 160b—annatto/bixin/norbixin (34%), sulphite (11%) and others (7%). Furthermore, the most numerous food additive notifications reported the presence of food additives in different spices, such as colour E 102—tartrazine, and colour E 124—Ponceau 4R in tandoori masala spices (8 notifications); colour E 160b—annatto/bixin/norbixin in paprika pepper (15 notifications) and chilli powder (16 notifications); sulphite unauthorised in garlic powder (16 notifications) and in cinnamon (16 notifications).

Finally, 100 notifications of adulteration-fraud were recorded of the following types: absence of health certificates and of certified analytical report (25%), absence of health certificates (52%), absence of certified analytical report (4%), improper health certificates (11%), missing import declaration (3%), illegal or unauthorised import (9%) and other adulteration-fraud (2%). These notifications, mainly the absence of health certificates and/or of certified analytical reports are frequent in chilli powder (38 notifications), curry leaves and curry powder (15 notifications), and spice mixes (12 notifications).

Most herbs and spices are subjected to different processes: cleaning, drying, disinfection, crushing, grinding, packaging, distribution, etc., and inappropriate handling and fraudulent practices can occur throughout the processing stage and in the food chain; these are difficult to detect in the final products that reach the consumer. Thus, in 2021, the European Commission launched a coordinated control plan, with the participation of 23 European countries, on the authenticity of herbs and spices [59]. This study analysed the purity of 1885 samples of herbs and spices marketed in Europe (pepper, oregano, saffron, turmeric, cumin and paprika) and found that 17% of the samples were adulterated. The most adulterated herb was oregano (48% of cases), with other plant species present: olive, marjoram, myrtle, mint, thyme, and sage. Among the spices, pepper had the highest percentage of adulteration (17% of samples), with rice starch, buckwheat, cereals, and mustard seeds detected. Cumin samples were adulterated with coriander, mustard, linseed, pumpkin seed, and caraway seeds. Some saffron samples contained other vegetables such as safflower and marigold, while turmeric samples contained paprika and corn, rice, and oat starch. In the case of paprika (adulterated in 6% of the samples), other vegetables were detected: corn, carrot, tomato, sunflower seeds, garlic, or onion. In addition, in the analysis of the spices saffron, paprika, turmeric and pepper, in 2% of 1340 samples the presence of synthetic colouring agents not authorized for human consumption was detected, with saffron and paprika being the spices with the highest rate of adulteration. The dyes detected were mainly Sudan I, colour E 102—tartrazine, colour E 129—Allura Red, colour E 122—azorubine, colour E 110—Sunset Yellow FCF, colour E 104—Quinoline Yellow and colour E120—Carminic Acid.

Table 4 shows detailed information on the notifications about potential risk encountered in herbs and spices in the period 2020–2022, showing a similar trend to that of the historical series analysed in relation to the type of adulteration [58].

In this 3-year period, 753 notifications were notified for herbs and spices, with 50 notifications (6.6%) being fraud and adulteration notifications. The number of notifications for each type of adulteration is as follows: novel food (2 notifications), allergens (11 notifications), composition (29 notifications), food additives and flavourings (7 notifications), labelling (1 alert) and adulteration-fraud (0 notifications). Furthermore, the risk decision was serious in 85% of these notifications. After analysis of the information, the notifications in unauthorised novel food and labelling were low and, in the adulteration-fraud category, null. In the allergen group, mainly celery, mustard, and/or gluten were detected in pure spices (basil, coriander, cumin, curry) and spice mix. In addition, unauthorized synthetic dyes continue to be the main protagonists of adulterations, mainly Sudan I (13 notifications) and Sudan IV (11 notifications) in curry, sumac, pepper, and other spices, as well as Orange II (9 notifications), mainly in chilli pepper.

The main notifying country for spice adulteration notifications was Belgium, followed by Spain and Switzerland. The origin of the adulterated spices varies according to the type of adulteration, notably Spain and India (in allergens) and Türkiye, Cameroon and Nigeria (in composition: illegal dyes). 

## 4. Scientific Articles Related to Fraud and Adulteration

A search for scientific articles related to herb and spice fraud was carried out. For this purpose, the Web of Science database [60] was used, using the keywords: herb* and spice* or herb* or spice*; and fraud*; and adulteration*; and authentication*. The results obtained, for the period 1990–2022, are shown in Figure 2. In this 32-year historical series, the total number of original scientific articles related to food fraud was 2334, of which 188 articles were related to “Fraud”, 879 articles to “Adulteration” and 1267 articles to “Authentication”. 

In the 1990s, only 21 articles were published, while in the 2000s and 2010s, 242 and 1507 articles were published, respectively. Therefore, there is an upward trend in research, mainly from 2005 to the present day, in the control of herb and spice fraud and in the use of increasingly efficient tools in the detection of different types of adulterants and evaluation of the authentication of food products.

In addition, the importance of fraud in certain herbs and spices can be seen from the remarkable number of research papers registered on the Web of Science. For example, the following records (number of papers per spice or herb), which include articles, book chapters and proceeding papers, stand out: >250 for sesame and saffron; >100 for turmeric and paprika; 50–100 for vanilla, oregano, thyme, black pepper, cinnamon, cumin, and cloves; 25–50 for chili, ginger, basil; and <25 for basil, anise, nutmeg, fennel, peppermint, and coriander, among others.

## 5. Methods for Detecting Spice and Herb Frauds and Adulterations

In this section, the different methods used to discriminate between adulterated and non-adulterated spices and herbs are discussed. In Table 5, an overview of some of the analytical methods for detecting frauds and adulterations is shown. In addition, an example of the different techniques applied to determine fraud or adulteration in paprika and chili powder, two of the most frequent spices adulterated according to RASFF databases, can be visualised in Figure 3. 

**Table 5 foods-12-03373-t005:** An overview of some important methods to detect spices/herbs frauds and adulterations.

Spices/Herbs	Fraud/Adulteration	Technique/Detection Method	Main Results	References
Paprika powder	Sudan I and Rhodamine B	HPLC-MS/MS	Detection of illegal synthetic dyes in Chinese paprika powder samples. Detection limits ranging from 0.013 ng/mL to 0.054 ng/mL, suggesting that the method is promising for accurate quantification of Sudan dyes at trace levels in foodstuffs.	[61]
Chili powder and paprika	Sudan I and II	HPLC-DAD	Detection of oil-soluble synthetic dyes in chilli products. The screening was based on the fingerprint differences of a normal unadulterated chilli sample with tested chilli samples. Limits of detection 0.40–2.41 mg/kg. The screening method was simple and had the possibility of finding the existence of the adulterated dyes which could not be identified using known standard analytes as control.	[62]
Chili powder	Sudan I–IV, Sudan Red 7B, Sudan Red G, Sudan Orange G, Para Red, and Methyl Red	UHPLC-DAD	Detection of the nine illegal dyes most frequently found in chilli-containing spices (the red dyes Sudan I–IV, Sudan Red 7B, Sudan Red G, Sudan Orange G, Para Red, and Methyl Re). Limits of detection showed lower values than required by European Union regulations and were in the range of 3.3–10.3 µg/L for standard solutions, and 5.6–235.6 µg kg^−1^ for chilli-containing spices.	[63]
Turmeric, curry, hot paprika, and sweet paprika	Synthetic dyes	HPLC-DAD	Very simple and fast detection of Sudan dyes (I, II, III and IV) in commercial spices up to a concentration of 5 mg/L.	[64]
Chili powder	Sudan	LC-UV/Vis	Simple detection of illegal dyes in foods such as Orange II, Sudan I–IV, Sudan Black B, Sudan Red 7B, Sudan Red G, Methanil Yellow, Dimethyl Yellow, Auramine O, Bixin, Fast Garnett GBC, Rhodamine B, Oil Orange SS, Orange G, Sudan Orange G, Naphthol Yellow, Acid Red 73, Toluidine Red, Sudan Red B, and Para Red in five matrices. The limits of detection, recovery and precision are considered adequate for a screening method.	[65]
Saffron	Geographical origin	HPLC-DAD	Differentiation of saffron spices produced at different sites on basis of crocin, safranal, picrocrocin and its derivatives and flavonoids. Statistical multivariate analysis of HPLC data offers the real possibility of differentiating PDO saffron from high-quality spices produced in close sites.	[66]
Saffron	Geographical origin	HPLC-DAD	Geographical discrimination of saffron samples from Iran and China on basis on picrocrocin and two types of crocin. The samples were well-separated according to their HPLC fingerprint data using PCA and orthogonal partial least squares discriminant analysis (OPLS-DA).	[67]
Saffron	Authenticity of saffron	HPLC-MS	Geographical discrimination of saffron samples on basis of glycerophospholipids and their oxidized lipids. The method allows for the distinguishing between PDO saffron and labelled Spanish saffron.	[68]
Paprika	Authenticate the geographical origin	HPLC-FLD	Phenolic acid and polyphenolic compounds were used as chemical markers to assess the classification of paprika from five European regions. The chromatographic fingerprints were also used to detect and quantitate two different paprika geographical-origin blend scenarios by partial least squares (PLS) regression.	[69]
Oregano	Olive leaves, myrtle leaves, cistus, hazelnut	LC-HRMS	Differentiation of oregano from olive leaves, myrtle leaves, cistus, and hazelnut by biomarker identification.	[37]
Oregano and sage	Olive leaves	GC-MS	Differentiation of ground oregano and sage from ground olive leaves on basis on two markers generated from the biophenol fraction. The detection limit was low, at 1%.	[70]
Bay leaves	*Cinnamomum tamala, Litsea glaucescens, Pimenta racemosa, Syzygium polyanthum* and *Umbellularia californica* leaves	GC-MS	Differentiation of bay leaf from its common surrogates (*Cinnamomum tamala, Litsea glaucescens, Pimenta racemosa, Syzygium polyanthum* and *Umbellularia californica* leaves).	[71]
Saffron	Turmeric and marigold	HS-GC-FID	Adulteration of saffron with two of the principal plant-derived adulterants: turmeric (*Curcuma longa* L.) and marigold (*Calendula officinalis* L.). The method, based on a combination of chemometrics with gas chromatography, may provide a rapid and low-cost screening method for the authentication of saffron.	[72]
Lemon balm	*Nepeta cataria* L.	CZE	Differentiation of the *Melissa officinales* L. from *Nepeta cataria* L. on basis on hidroxycinnamic acid contents for detection of commercial substitutions.	[73]
Vanilla	Artificial flavourings Adulterations	CZE)	Identification of natural vanilla by detection of p-hydroxybenzaldehyde, p-hydroxybenzoic acid, vanillin, and vanillic acid; identification of artificial flavourings by detection of ethyl vanillin; identification of adulterations by detection of coumarin in vanilla samples. Limits of detection from 2 to 5 μg/mL.	[74]
Smoked paprika	Adulteration with non-smoked paprika from non-authorized varieties	CZE	Detection of frauds in smoked paprika POD “Pimentón de La Vera” by mixing with non-authorized varieties. Methanol soluble proteins and hidrophilic and hidrophobic protein fractions allowed the detection limit of 5%.	[75,76]
Saffron	*Curcuma* species	RAPD-PCR	Four RAPD primers (OPA 02, OPA 04, OPA 07, and OPC 05) were used. RAPD banding pattern of two *Curcuma* species, namely *Curcuma longa* L. and *Curcuma zedoaria* (Christm.) Roscoe, and three market samples were tested to evaluate adulteration. Three market samples of turmeric powder were adulterated with *Curcuma zeadoaria* (Christm.) Roscoe.	[77]
Chili powder	Dried red beet powder, almond shell dust and powdered *Ziziphus nummularia* fruits	RAPD-PCR	Three selected RAPD primers (OPA-2, OPA-15 and OPA10) which produced adulterant-specific bands in simulated samples were used for analysing market samples of chilli powder. Out of the six market samples analysed, one sample showed an amplified *Ziziphus nummularia*-specific band, indicating the occurrence of adulteration in market samples. All the market samples tested were free from dried red beet pulp or almond dust adulteration.	[78]
Black pepper	*Carica papaya*	RAPD-PCR	Five decamer oligonucleotide primers (OPC-1, OPC-4, OPC-6, OPC-7 and OPC-8) discriminated *Piper nigrum*, as well as *Carica papaya*, by the presence and absence of unique bands.	[79]
Smoked paprika P.D.O. “Pimentón de la Vera” (autochthonous varieties of pepper: *Jaranda, Jariza*, and *Bola*)	Paprika elaborated from varieties of pepper foreign to the La Vera region, in central western Spain (varieties: *Papri Queen, Papri King, Sonora, PS9794,* and *Papri Ace*)	RAPD-PCR	RAPD-PCR with primers S13 and S22: two molecular markers of 641 and 704 bp, respectively, were obtained, which allowed all of the smoked paprika varieties to be differentiated from paprikas elaborated with the five foreign varieties.	[80]
Oregano	Plants lacking a clearly detectable essential-oil profile (*Rubus* sp., *Cistus incanus* L., *Rhus coriaria* L)	RAPD-PCR	Thirteen RAPD primers discriminated between oregano and its adulterants, allowing their detection in oregano samples with a limit of detection of 1%.	[81]
Oregano	*Cistus incanus* L., *Rubus caesius* L., and *Rhus coriaria* L.	SCAR-PCR	Detection limits at 1% for the adulteration of oregano.	[82]
Oregano	Olive leaves	SCAR-PCR	Detection limits at 1% for the adulteration of oregano.	[83]
Saffron	*Curcuma* species	SCAR-PCR	Two pairs of SCAR primers were designed from the RAPD markers ‘Cur 01’ and ‘Cur 02’, respectively. Six market samples of turmeric powder and four simulated standards besides the genuine samples were analysed using the specific SCAR markers. Both the SCAR markers detected the presence of *Curcuma zedoaria*/*Curcuma malabarica* adulteration in four market samples and inall the simulated standards prepared in different concentrations. The efficiency of the SCAR markers for detecting adulteration even at low concentrations (10 g adulterant/kg of turmeric powder) substantiates their applicability as a qualitative diagnostic tool for detecting plant-based adulterants in turmeric powder.	[84]
Chili powder	Dried red beet pulp and powdered *Ziziphus mummularia* fruits	SCAR-PCR	Red beet pulp-specific SCAR primer pair, B1, and *Ziziphus nummularia*-specific SCAR primer pair, Z1, were designed from the corresponding RAPD marker sequences to amplify SCAR markers of 320 bp and 389 bp, respectively. SCAR markers could detect the adulterants at a concentration as low as 10 g adulterant kg/blended sample. The *Z. nummularia* SCAR marker could detect the presence of *Z. nummularia* fruit adulteration in one of the commercial samples. All the market samples tested were free from red beet pulp adulteration.	[51]
Saffron	Safflower and *Calendula*	SCAR-PCR and DNA barcoding	SCAR markers SAFL4, SAFL40, SCCt131, and ScCO390 were useful for simple, accurate, specific, and sensitive detection of safflower/*Calendula* adulteration in saffron. Out of the three DNA barcodes (psbA-trnH, ITS2, and rbcLa) used, psbA-trnH was considered ideal for detection of adulterants in saffron as it gave different product sizes for saffron and safflower/Calendula. Detection limits of safflower (0.5%) and *Calendula* (3%) in saffron.	[85]
Saffron	Safflower, corn	SCAR and ITS multiplex PCR-based assay	Six pairs of SCAR primers were designed which were able to amplify reproducible saffron DNA with expected sizes and no amplification in corn and safflower DNA. In this study, a primer pair was also designed based on ITS sequences for specific amplification of safflower DNA. PCR reactions specifically amplified 613 bp of ITS region in safflower genome. The multiplex PCR assays were further established for the joint use of some SCAR and ITS markers efficiently.	[86]
“Florinis” Greek pepper	Florinis-type pepper and Karatzova peppers	ISSR	Differentiation of “Florinis Greek” pepper from its adulterants. The economic interest in ‘Florinis’ peppers has led to many adulteration events. In that aspect, genetic profiles of ‘Florinis’, a ‘Florinis’-type and ‘Karatzova’ peppers, were studied using inter-simple sequence repeat (ISSR) molecular markers and an automated fragment detection system. The molecular protocol established during this study may successfully discriminate the original ‘Florinis’ cultivar from the ‘Florinis’-type peppers and ‘Karatzova’ cultivar.	[87]
Oregano	Adulterants	SSR	Simple sequence repeat (SSR) markers were developed from expressed sequence tags (ESTs) of essential oil glands of oregano. Thirteen EST-SSR loci were evaluated using 20 individual plants of oregano and 19 plants of *Origanum majorana*.	[88]
Saffron	Safflower	SCAR-RAPD	Identification of the adulterant (safflower petals) in commercial saffron samples by amplification of two specific bands by SCAR primers designed from RAPD bands. Limit of detection: 1% of safflower.	[89]
Turmeric powder	Cassava, wheat, barley, rye starches	DNA barcoding	*ITS* was the ideal locus among the three testes (rbcL, ITS and matk) to discriminate the *Curcuma* species. Adulterants including *Curcuma zedoaria* (in one sample) and cassava starch, wheat, barley, and rye (in other two samples).	[90]
Cumin, garlic, fennel, cinnamon, pepper, bay leaves, clove	Wheat, sorghum, maize, soybean, rice species	DNA barcoding	A total of 22 species (16 types of spices and 6 adulterations) were collected for this study. ITS2 and psbA-trnH were used as barcoding loci. Only two types of natural spices (fennel and liquorice) were correctly labelled; the other 14 spices had different amounts of adulteration.	[91]
Sixteen types of culinary spices from Beijing Tong Ren Tang Group. Coriander, bay leaf, white pepper, and cumin	*Triticum aestivum* (wheat), *Oryza sativa* (rice) and *Zea mays* (maize)	DNA barcoding	Evaluation of five barcodes (ITS2, rbcL, trnL (UAA), trnL (P6 Loop), and psbA-trnH). Combination of two barcodes (ITS and psbA-trnH) gave a higher species’ resolution rate (95.5%). Thirty commercial products were evaluated, with 93.3% of the tested products being authentic and 6.7% indicating adulteration with rice.	[92]
Basil, oregano, paprika	Wind-pollinated plant species, wind-spread plant species	DNA metabarcoding	In this study, DNA metabarcoding was used for the identification and authentication of 62 products containing basil, oregano, and paprika, collected from different retailers and importers in Norway. Results showed varying degrees of discrepancy between the constituent species and those listed on the product labels, despite high product authenticity.	[93]
Saffron	*Daucus carota*, *Carthamus tinctorius*, *Calendula officinalis*, *Dendranthema morifolium (Ramat.) Tzvel*., *Nelumbo nucifera*, *Hemerocallis fulva* (L.) L., and *Zea mays*	Barcoding melting curve analysis method (Bar-MCA)	The universal chloroplast plant DNA barcoding region trnH-psbA was used to identify adulterants of saffron. Differences between the melting temperatures of saffron and its adulterants can be used to discriminate authentic and adulterated saffron.	[94]
Spices from *Lamiaceae* family	Adulterants	DNA barcoding	The barcode regions (rpoB, rbcL, matK and trnH-psbA) were tested. Results suggest that the non-coding trnH-psbA intergenic spacer was the most suitable marker for molecular spice identification, followed by matK. Both markers were almost invariably able to distinguish spice species from closest taxa, with the exclusion of samples belonging to the genus Oregano.	[95]
Saffron	Adulterants	Real-time PCR + HRM analysis and DNA mini-barcodes	ITS1 and matK region markers for *Crocus* genus detection. ITS2 locus for species-specific detection of *Crocus sativus* and *Crocus cartwrightiamus*	[96]
Turmeric powder	Sudan Red, starch, and Metanil Yellow	NIR spectroscopy	Controlled (PCA) and uncontrolled (PLS-DA and CMCA) pattern-recognition techniques for the detection and classification of Sudan Red, starch and Metanil Yellow fraud were applied to spectra. The overall precision of the SIMCA and PLS-DA classifiers were 82% and 92%, respectively	[97]
White pepper	Corn flour	1. Portable NIR spectrometer 2. Hyperspectral imaging	1. Recognition models by LDA, SVM, PLS-DA and SIMCA. The SIMCA model performed best in quality grading. For optimized PLS model on piperine concentration, prediction of unknown samples generated an R^2^_p_ of 0.970, RMSEP of 0.111, and RPD value of 5.72. 2. The MCR-ALS was used to reduce the dimensionality of multivariate data. Minimum adulteration content of 1%.	[98]
Ginger powder	Chickpea powder	Hyperspectral imaging	Recognition models by convolutional neural networks (CNN). CNN was able to grade the images of ginger powder with 99.70% accuracy, compared to other classifiers.	[99]
Nutmeg	Seven adulterant materials: pericarp, two creamy spent, three brown spent, and one shell	Hyperspectral imaging	Data were pre-processed using standard normal variate (SNV) treatment. An artificial neural network (ANN) model showed the ability to detect adulteration at levels as low as 5%.	[100]
Cinnamon	*Cinnamon cassia* (10, 50, and 100%)	NIR spectroscopy	Average discrimination percentages of 99.25 and 100.00% for recognition models PLS-DA and probabilistic neural network (PNN), respectively.	[101]
Cinnamon	*Cinnamon cassia*	Hyperspectral imaging	PLS-DA and support vector machine (SVM) reached a similar performance to classify samples according to origin, with error = 3.3% and accuracy = 96.7%.	[102]
Turmeric	Starch	FT-NIR spectroscopy	Wavelength regions selected: 1400–1550 nm and 1900–2050 nm by variable importance in projection (VIP) method. PLSR model (R^2^ > 0.91).	[103]
Ginger	Corn starch, soybean flour, and wheat flour.	FT-NIR spectroscopy	Random forest (RF) and gradient boosting (GB) algorithms exhibited the highest accuracies (100%) in classification. PLSR models were built to further determine whether the adulterated levels of ginger adulteration with RPD values are greater than three for the three adulterants.	[104]
Saffron	Plant-derived adulterants	FT-NIR spectroscopy	PLS-DA on region 4000–600 cm^−1^ (99% correct classification of pure saffron and saffron adulterated at 5–20%). Synergy interval PLS (siPLS) with detection limits ranging from 1.0% to 3.1%.	[27]
Turmeric	Rice flour with tartrazine	Hyperspectral imaging	Functional relationship between the Bhattacharyya distance and the adulteration levels. Multivariate Gaussian. Model (R^2^ = 0.9816 and SSE = 1.1423).	[105]
Turmeric	Metanil Yellow, Sudan I	Raman spectroscopy	Self-modelling mixture analysis (SMA) was used to decompose the mixed spectral information. Linear correlation (R^2^ = 0.99).	[106]
Chili powder	Use of rhodamine B as a synthetic colourant	Indirect competitive ELISA	Immunoassay strategy was designed based on the heterologous strategy. Detection limit of indirect competitive ELISA was 0.002 μg/kg, showing a good sensitivity.	[107]
Chili	Sudan I, as a colorant	ELISA	Development of rapid ELISA method based on highly specific polyclonal antibodies. This ELISA method allows rapid, sensitive, and high-throughput screening of different food products for the presence of the illegal colorant.	[108]
Chili, curry, and mixes of soup and condiment	Detection of gluten-free product	ELISA kits	These authors detected levels exceeding the gluten threshold (20 ppm) in some of the condiment samples tested.	[50]
Cumin spices	Detection of traces of peanut protein	ELISA kits	A lack of sensitivity of the ELISA kit for traces of peanut protein in cumin spices because of false negative results.	[48]
Turmeric	Adulteration with other curcumin pigment	Light and scanning-electron microscopy	The turmeric powder can be identified by the presence of gelatinized starch granules, numerous oil cells, and parenchymatous cells in a microscopic view. The presence of calcium oxalate crystals in turmeric indicates adulteration with wild species.	[22]
Cumin, chilli, pepper and mustard powders	Adulterating substances (starch, plant straws, and monosodium glutamate)	Microscopic technique	Showed the efficiency of using microscopic technique to distinguish the micro-morphology characteristics of pure seasoning powders.	[109]
Black pepper powder	Adulteration with papaya seed powder	Microscopic technique	Meticulous microscopic examination of fatty oils, oil globules, starch granule, fibres and different features of parenchyma cells identified papaya seed powder in black pepper powder.	[110]
Fennel	Combined microscopy and GC-MS for the detection of adulteration of fennel seeds	Combined microscopy and GC-MS	Combined light microscopy coupled with fluorescence microscopy and GC-MS analysis allowed successful distinguishing of fennel seeds from two adulterants: dill (*Anethum graveolens*) and cumin (*Cuminum cyminum*).	[111]
Saffron	Adulterated with different percentages of dyes	Electronic nose or E-nose and a chemometric tool	The results of the analysis revealed that E-nose and a chemometric tool were able to differentiate authentic saffron samples from adulterated ones effectively, based on their aroma intensity.	[112]
Cumin	Adulterated with Moroccan coriander in different concentrations (5%, 20%, 50% and 70%)	E-nose and VE-tongue in combination with SPME-GC-MS	Compared the ability of E-nose and VE-tongue in combination with SPME-GC-MS to discriminate cumin samples from adulterated ones and those with different geographical origins; this was demonstrated. The results indicated that the VE-tongue has more potential (100% accuracy) for detection and discrimination than the other two methods.	[113]
Saffron	Safflower and corn-stigma adulteration in saffron	Electronic nose	The results revealed that the system can successfully recognise saffron adulteration with 100 % accuracy, and that it was able to successfully differentiate unadulterated saffron from adulterated saffron, with an adulteration level of more than 10%.	[114]

**Figure 3 foods-12-03373-f003:**
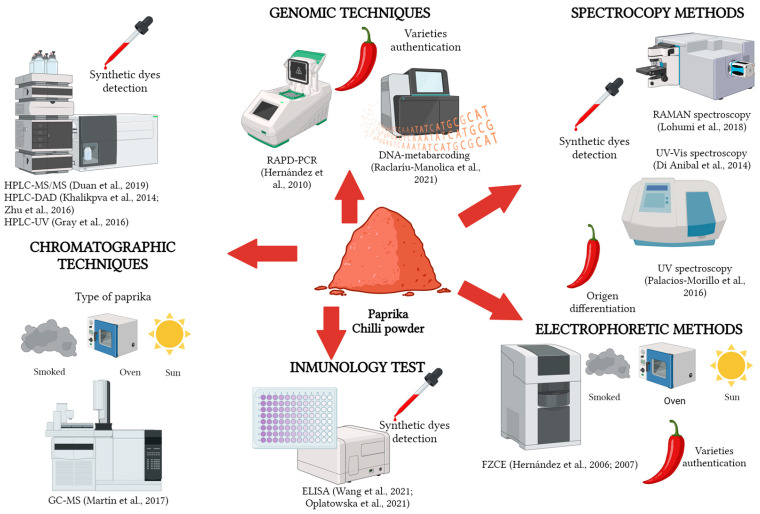
Overview of different methods to detect paprika and chili-powder fraud and adulteration [61,62,63,65,75,76,80,93,107,108,115,116,117,118]. Created with BioRender.com.

### 5.1. Chromatography Techniques 

Chromatographic techniques have great applicability in the resolution of food analytes. One of the most interesting applications is analysing the authenticity of food, including herbs and spices. For fraud and adulteration detection, these techniques include thin-layer chromatography (TLC), high-performance liquid chromatography (HPLC), gas chromatography (GC), and methods based on mass spectrometry (MS) (liquid chromatography (LC)-MS/MS, GC/MSD) [119]. TLC is a method that has been widely used for the analysis of natural and synthetic dyes due to its advantages. TLC is a simple technique with numerous detection possibilities and low operational costs [120]. However, TLC, compared to other techniques, is not an efficient technique in detection applications. MS allows quantifying known analytes at very low concentrations; it is a highly specific and sensitive technique, although it is expensive and requires significant laboratory expertise [119,121]. It is a powerful tool in the fight against food fraud, and in many industries it is considered the gold-standard technique [25].

HPLC is a useful, reliable, and powerful technique for detecting adulterations and fraud in spices and herbs, as it allows the identification and quantification of several compounds in a single measurement. It is highly sensitive, robust, cost-effective, and reproducible. However, this technique usually involves many extraction and purification steps, which are considered bottlenecks in analytical methods [122]. The detectors used in HPLC are various: ultraviolet–visible (UV–Vis), fluorescence (FLD), diode array detector (DAD), refractive index (IR) and (MS) detectors.

HPLC has allowed the detection of synthetic dyes in spices. Duan et al. [61] analysed fifteen synthetic dyes in Chinese paprika powder samples using HPLC-MS/MS and detected that seven of the analysed samples contained illegal synthetic dyes such as Sudan I and Rhodamine B. This chromatographic technique and/or LC, together with detectors such as DAD and UV–Vis, is also used for the detection of adulterated spices with synthetic dyes such as chili powder, paprika and turmeric. Zhu et al. [62] developed an HPLC-DAD method for the simultaneous identification and quantification of six synthetic dyes in samples of chili powder and paprika. They found that some of them were adulterated with Sudan I and II. Adulteration of chili powder with nine illegal dyes, most frequently found in chilli-containing spices (Sudan I–IV, Sudan Red 7B, Sudan Red G, Sudan Orange G, Para Red, and Methyl Re), was also detected using UHPLC-DAD [63]. Other authors analysed the adulteration with Sudan dyes in 27 samples of commercial spices (turmeric, curry, hot paprika, and sweet paprika) using chromatography (HPLC-DAD), obtaining good and satisfactory results [64]. Other authors were able to detect the presence of Sudan dye in chili powder using LC-UV/Vis [65]. In other research, HPLC with different detectors for identifying other analytes and the authenticity of spices according to their geographical origin has been employed. D’Archivio et al. [66] analysed saffron samples from different regions of Italy using HPLC-DAD. Compounds such as crocin, picrocrocin, and flavonoids allowed for the differentiation of the samples from each analysed region. Similarly, other authors discriminated Chinese and Iranian saffron samples by identifying flavonoids, picrocrocin, and different types of crocin using HPLC-DAD [67]. Significant markers in saffron, such as glycerophospholipids and their oxidized lipids, have been identified using HPLC-MS for this product authentication [68]. Campmajó et al. [69] identified phenolic acid and polyphenolic compounds, as a chemical marker to authenticate the geographical origin of paprika using HPLC-FLD. Black et al. [37] developed and validated LC–high-resolution mass spectrometry (LC-HRMS) to screen for and confirm oregano adulteration with olive leaves, myrtle leaves, hazelnut leaves, sumac leaves, and cistus leaves. This methodology allowed the identification of 16 unique markers in positive mode and 12 in negative mode, with all adulterant samples having at least 4 unique markers.

Gas chromatography (GC) is another method that has been used to detect possible adulterants in spices and herbs, mainly to analyse volatile and semivolatile substances and aromatic compounds [123]. The detectors utilized with this chromatographic technique are the thermal conductivity detector (TCD), flame ionization detector (FID) and mass spectrometer (MS). These methods have the advantages of ease of use, allowing for the precise identification of compounds even in complex samples, and for result reproducibility. Therefore, the addition of volatile oils to a specific herb or spice can mislead the GC-MS adulteration-detection method [25]. Therefore, the addition of volatile oils to a product may cheat the GC-MS adulteration detection method [25]. Bononi, et al. [70] used GC-MS to detect olive leaves in oregano and sage. Raman et al. [71] were able to accurately distinguish bay leaves from similar leaves of other species such as *Cinnamomum tamala* (Nees & Eber), *Litsea glaucescens* (Spreng. ex Nees), *Pimenta racemosa* (Mill.) J.W.Moore, *Syzygium polyanthum* Thwaites and *Umbellularia californica* (Hook. & Arn.) Nutt. Fatty acid profiles allowed them to identify adulterations with palmitic, palmitoleic, stearic, and myristic acids. Martín et al. [115] utilized a GC-MS method capable of discriminating between the volatile compound profiles associated with smoked paprika (which is under PDO “Pimentón de La Vera”) and other profiles linked to other kinds of drying that are not under the abovementioned PDO (oven-dried and sun-dried paprikas). Another method to detect adulterations based on the volatility of analytes is GC with FID. Morozzi et al. [72] used a non-targeted approach based on the combination of headspace flash gas-chromatography with flame ionization detection (HS-GC-FID) and chemometrics to check the adulteration of saffron with two of the principal plant-derived adulterants: turmeric (*C. longa* L.) and marigold (*Calendula officinalis* L.).

### 5.2. Electrophoretic Methods 

Capillary electrophoresis comprises a highly versatile group of analytical techniques relying on the separation of molecules on the basis of electrophoretic mobility, molecular weight, electrical charge, or a combination of all three. Capillary electrophoresis is a rapid and sensitive technique for the detection of a wide range of analytes, with advantages over other instrumental techniques in terms of the volume of sample required and the lower cost of solvents.

There are different modalities of capillary electrophoresis, among which the following stand out: capillary zone electrophoresis (CZE), which bases its separation on differences in charge/mass ratios; micellar electrokinetic chromatography (MEKC), in which the addition of detergents allows the generation of micelles; capillary gel electrophoresis (CGE), in which the separation occurs inside a capillary filled with a gel that acts as a molecular sieve; and isoelectric focusing (IEF), where molecules migrate under the influence of the electric field, provided they are charged, in a pH gradient. Depending on the extraction method applied and the variant of capillary electrophoresis used, this group of techniques has been employed in the separation of proteins and peptides, nucleic acids, organic acids, polyphenols, etc.

The different modalities of capillary electrophoresis have been widely used for the characterisation of bioactive compounds in spices and herbs [124]. In the last 10 years, several studies have been carried out to develop analytical methods for these metabolites. Maher et al. [125] developed a method for the detection of luteolin and apigenin in thyme and parsley, using an electrolyte solution of borax and methanol, while Głowacki et al. [126] used MEKC for apigenin analysis. In the same way, methods for the analysis of bioactive curcuminoids from turmeric herbal products have been developed using a non-aqueous background electrolyte [127], borate [128], and 20 mM Triton X-100, 20 mM SDS, 30% (*v*/*v*) methanol in 10 mM borax solution at pH 10.0 [129]. Other capillary electrophoresis methods for other compounds such as rosmarinic acid and carnosic acid from *Salvia* species [130], and flavonoids and glycosides from *Vitex negundo* L. [131] have been developed.

In the area of fraud and adulteration, the detection of a phenolic compound characteristic of *Melissa oficcinalis* L. such as 3-acetylcoumarin, 6-hydroxycoumarin, cinnamic acid, 4-hydroxycoumarin, 4-hydroxycinnamic acid, rosmarinic acid and caffeic acid has allowed the detection of fraud related to the false marketing of *Nepeta cataria* L. as *M. oficcinalis* L. [73].

The application of electrophoresis to discern between natural and artificial flavourings and potential adulterations of vanilla can also be cited. In other method [74], a tetraborate buffer + 20% ethanol and diode array detector were used. The detection of ethyl vanillin is associated with artificial flavours, while the presence of coumarin is associated with vanilla adulterations.

Saffron is one of the spices that is the subject of a large number of analytical techniques developed to guarantee its authenticity. However, the bibliography consulted only shows the development of a capillary electrophoresis method for the quantification of vitamin B2, using a borate buffer [132].

The analysis of different protein fractions by capillary electrophoresis using phosphate buffers has allowed the detection of fraud associated with the marketing of smoked paprika under the PDO “Pimentón de La Vera”. Methanol-soluble protein fractions [75] and protein partitioning using the detergent Triton X-100 [76] allowed the development of a sensitive method to detect non-permitted pepper varieties and unauthorised drying methods.

In addition to methods developed to analyse phytochemicals, volatiles, and protein fractions, the resolution of molecular techniques by capillary electrophoresis has been used for the detection of fraud in spices and herbs. As an example, we can mention the authentication of herbal teas by DNA barcoding of Plastid Noncoding DNA [133].

### 5.3. Genomic Techniques 

Although there is an important rise in the development of emerging non-destructive techniques, DNA-based techniques are increasingly used to detect fraud and adulteration in spices and herbs. These methods are less expensive, more efficient, and more accurate than others, so they are a good instrument against fraud. Among these types of techniques, random amplified polymorphic DNA (RAPD), sequence-characterized amplified region–polymerase chain reaction (SCAR-PCR), an advancement on the RAPD markers in DNA analysis, and DNA barcoding are the most popular, and are becoming desirable methods for the detection of spice frauds and adulteration. Although other DNA-based techniques variants such as inter-sequence simple repeat (ISSR), simple sequence repeats (SSRs), or, more recently, the combination of DNA barcoding with high-resolution melting (HRM) have been also employed.

SSRs are tandem repeats of simple sequences, consisting most frequently of two, three or four nucleotides that can be repeated 10–100 times. The copy number of these repeats, which can be highly variable due to unequal crossing over, is the basis for the polymorphism [134]. This technique was used by Novak et al. [88] to develop markers able to differentiate oregano (*O. vulgare* L.) from its adulterants. Recently, Mougiou et al. [87] utilized the ISSR technique, which is a PCR-based method which uses microsatellites as primers in a single reaction, targeting multiple genomic loci, to successfully discriminate ‘*Florinis*’ Greek pepper from its adulterants (‘*Florinis*’-type and ‘*Karatzova*’ peppers).

RAPD-PCR markers are DNA fragments from PCR amplification of random segments of genomic DNA with single primer of arbitrary nucleotide sequences. This DNA-based technique has been widely used for the identification of plant-based adulterants in spices and herbs in the last fifteen years [77,79,81,84]. Sasikumar et al. [77] optimized a method using four RAPD primers (OPA 02, OPA 04, OPA 07, and OPC 05) to detect two *Curcuma* species, namely *C. longa* L. and *Curcuma zedoaria* Roscoe, contaminants of turmeric powder. They found that three market samples of turmeric powder were adulterated with *C. zeadoaria*. Dhanya et al. [78] developed a method consisting of three selected RAPD primers (OPA-2, OPA-15 and OPA10) which produced adulterant specific-bands from dried red beet powder, almond shell dust and powdered *Ziziphus nummularia* (Burm. f.) Wight & Arn. fruits to discern the authenticity of chili powder. Results showed that out of the six market samples analysed, only one sample amplified the *Z. nummularia*-specific band, indicating the occurrence of adulteration in market samples with this adulterant. Khan et al. [79] carried out a method based on five decamer oligonucleotide primers (OPC-1, OPC-4, OPC-6, OPC-7 and OPC-8) capable of discriminating the spice black pepper (*P. nigrum* L.) and its adulterant *C. papaya* L. by the presence and absence of unique bands. Marieschi et al. [81] designed a method that relied on thirteen RAPD primers discriminating between oregano (*Origanum* spp.) and its adulterants which lack a clearly detectable essential oil profile (*Rubus caesius* L., *Rhus coriaria* L. and *Cistus incanus* L.). This method allowed the determination of oregano sample adulteration with a limit of detection of 1%. Finally, Hernández et al. [80] optimized a RAPD-PCR with primers S13 and S22 that allowed the obtaining of two DNA markers to distinguish between smoked paprika varieties protected by the Spanish PDO, “Pimentón de La Vera” (*Jaranda, Jariza,* and *Bola*), and paprikas elaborated with the five foreign pepper varieties (*Papri Queen*, *Papri King*, *Sonora*, *PS9794*, and *Papri Ace*).

SCAR-PCR (Sequence Characterized Amplified Region) is a technique based on RAPD-PCR that increased the sensitivity, specificity and reliability of the basic technique and is used for genetic characterization and authentication of organisms. SCAR marker is a DNA fragment amplified by PCR using specific 18–26 bp primers; these primers link to traits of interest and are designed from nucleotide sequences cloned from RAPD fragments [135] or other DNA marker-derived fragments. This DNA-based technique has been widely used in the last decade for the detection of adulterations and/or fraud in spices. Thus, Marieschi et al. [82,83] developed two SCAR-PCR methods to determine adulteration of oregano with *C. incanus* L., *R. caesius* L., and *R. coriaria* L., and olive leaves, respectively. Both methods were designed from a previous RAPD-PCR method [81] and permitted the determination of oregano sample adulteration with a limit of detection of 1%. Another SCAR-PCR method to identify adulteration of chili powder with dried red beet pulp and powdered *Z. mummularia* (Burm. f.) Wight & Arn. fruits was carried out [51]. SCAR primers were specifically designed to detect both adulterants from the corresponding RAPD marker sequences to amplify SCAR markers of 320 bp and 389 bp for dried red beet pulp and powdered *Z. mummularia* fruits, respectively. This method could detect the adulterants at a concentration as low as 10 g adulterant kg/sample and it was used to demonstrate the adulteration of market samples with powdered *Z. mummularia* fruits. Although this technique has been used for determination of the authenticity of different spices, it has been more employed to determine the adulteration of saffron, an expensive and valuable spice worldwide. Two SCAR-PCR methods have been designed to identify adulteration of saffron with *Curcuma* species (containing the colouring pigment curcumin, such as *C. zedoaria* Roscoe and *Curcuma malabarica* Velay., (Amalraj & Mural.) or safflower [84,89]. Dhanya et al. [84] designed two pairs of SCAR primers from the RAPD markers ‘Cur 01’ and ‘Cur 02’, to detect *C. zedoaria* and *C. malabarica*, saffron adulterants, respectively. The efficiency of the SCAR markers was high, since they were able to detect adulteration even at low concentrations (10 g adulterant/kg of turmeric powder). Javanmardi et al. [89] identified the adulterant (safflower petals) in commercial saffron samples by the amplification of two specific bands by SCAR primers, with the limit of detection of 1% of safflower. In addition, Babaei et al. [86] developed a SCAR and ITS multiplex PCR-based assay that consisted of developing SCAR primers specific for saffron DNA amplification and ITS primers specific for the amplification of its adulterant (safflower). It is obvious that SCAR-PCR is a sensitive technique and is quite useful for detection of spice fraud/adulteration, but there is a need for sequence data for the primer design.

DNA barcoding is based on the conservation of DNA fragments within a species and variability between species. These are the genetic markers of barcodes. This technique, alone or in combination with others, has been extensively used in the detection of adulterants in herbs and spices. De Mattia et al. [95] developed a method to discriminate spices belonging to the Lamiaceae family (*Mentha*, *Ocimum*, *Origanum*, *Salvia*, *Thymus* and *Rosmarinus*) from their adulterants. For this, they evaluated four barcode regions (alone and in combination), and they suggested that the non-coding trnH-psbA intergenic spacer is the most suitable marker for molecular spice identification. In addition, they proposed a multilocus barcode approach based on the combination matK+trhH-psbA to distinguish between the target spices and their adulterant. Markers were almost invariably able to distinguish spice species from the closest taxa, with the exclusion of samples belonging to the genus *Oregano*. Zhang et al. [91] employed this technique to investigate the adulteration of 16 types of powdered natural spices (cumin, garlic, fennel, cinnamon, pepper, bay leaves, and clove, among others) with congeneric species; cheaper substitutes with a similar colour or appearance; or crop-based products such as rice, corn, or wheat flour. For this, ITS2 and psbA-trnH were used as barcoding loci, and they found that only two types of natural spices (fennel and liquorice) were correctly libelled; the other 14 spices had different amounts of adulteration. More recently, Zhou et al. [92] demonstrated that a combination of two barcodes (ITS and psbA-trnH) previously identified as optimum by Zhang et al. [91] permitted the evaluation of adulteration of 16 types of culinary spices from the Beijing Tong Ren Tang Group with *Triticum aestivum* L. (wheat), *Oryza sativa* L. (rice) and *Zea mays* L. (maize). They observed that two out of thirty commercial products tested were adulterated with *O. sativa*. Raclaríu-Manolica et al. [93] proposed the use of DNA metabarcoding in combination with appropriate traditional chemical methods to detect the adulteration of basil, oregano and paprika. They highlighted the necessity for proper analytical validation of DNA metabarcoding before it can be implemented for molecular diagnostics. On the other hand, the high economic value of saffron induces traders to adulterate it; therefore, numerous DNA barcoding methods have been developed to trade while maintaining the authenticity of this spice. Parvathy et al. [90] designed a method to detect *C. zedoaria* and cassava, wheat, barley, and rye starches as adulterants of this valuable spice. They suggested that ITS was the ideal locus among the three tested (rbcL, ITS, and matk) for discriminating the *Curcuma* species. Adulterants including *C. zedoaria* (in one sample) and cassava starch, wheat, barley, and rye (in two samples) were encountered. Bansal et al. [85] optimized a SCAR-PCR and DNA barcoding method. They demonstrated that the SCAR markers SAFL4, SAFL40, SCCt131, and ScCO390 and the barcoding locus psbA-trnH were useful for the detection of safflower/*Calendula* adulteration in saffron. The detection limits were 0.5 % (safflower) and 3% (*Calendula*) in saffron (*C. sativus* L.). Jiang et al. [94] and Villa et al. [96] designed DNA barcoding methods in combination with HRM analyses to detect adulteration in saffron. Jiang et al. [94] utilized the barcoding region trnH-psbA to identify adulterants, while Villa et al. [96] employed ITS2 for the species-specific detection of *C. sativus*. Both demonstrated that the melting temperatures of adulterants differed from that obtained for saffron.

### 5.4. Spectroscopy and Image Analysis Methods

Spectroscopy methods are based on the study of the interaction between electromagnetic radiation and matter’s structure and composition, with absorption or emission of radiant energy. The use of these methods for adulterant authentication in spices and derivates includes ultraviolet and visible (UV–vis), infrared, vibrational, fluorescence, Raman, MS, and nuclear magnetic resonance (NMR). Ultimately, these modern instrumental techniques have been mostly combined with statistical tools based on univariate and multivariate (chemometrics) statistics for the detection of adulterations and frauds in spices and herbs.

UV–vis spectroscopy is a simple and cheap method based on the absorbance of chemical groups such as aromatic, conjugated, or unsaturated compounds, requiring in most cases a pre-treatment of a sample. Despite the difficulty for UV–vis spectrum interpretation due to its complex nature, this technique has been widely utilized, in combination with multivariate analysis methods, to identify and quantify artificial colorants in spices and herbs. Di Anibal et al. [116] detected adulteration up to 1–5 ppm of Sudan I and blends of Sudan I + IV dyes from UV–vis spectra of ethanolic extracts of three varieties of paprika, using as classification techniques partial least squares discriminant analysis (PLS-DA) and k-nearest neighbors (KNN). The usefulness of the PLS-DA and UV–vis spectrometry combination for identifying banned Sudan dyes in commercial spices at the referenced concentration level, both individually and in mixtures of different proportions, was corroborated [136]. The analysis of second derivative UV–vis spectra of saffron aqueous extracts was effective to detect adulteration of this spice with carminic acid down to the level of 2.0% (*w/w)* [137]. UV spectroscopy has been also used for differentiation and geographical classification of spices. Differentiation of paprika from the two Spanish PDOs, Murcia and Extremadura, were carried out by a pattern-recognition classification model of UV–vis spectra of paprika acetonic extracts using multilayer perceptron artificial neural networks (MLP-ANN) [117]. Linear discriminant analysis (LDA) based on intensity of UV–visible spectra of saffron aqueous extracts provided a correct geographical classification of samples from four different production areas [138]. However, the need to use transparent and pre-treated samples with a limited complexity determines the utility of the UV–vis spectrometry as a routine technique for determining the authenticity of herbs and spices.

According to the recent reviews [139,140,141], IR-spectroscopic techniques seem to be most useful to analyse and authenticate spice samples for qualitative and quantitative analysis, in combination with multivariate chemical analysis. Based on the frequency of radiation applied, IR spectroscopy can be categorised into Near-IR (NIR; 750–1400 nm) and mid-IR (MIR; 1400–25,000 nm). Today, for most research and development-grade mid-IR instruments, the signal is subjected to the Fourier transform function using an interferometer to generate the spectrum (FT-IR). Samples for FT-IR analysis are solid, with sifted treatment. The adulteration of black pepper and cumin powder with cassava starch and corn flour was efficiently detected by mid-IR, using for the analysis of spectra the multivariate technique SIMCA (soft independent modelling of class analogy), among others [142]. FT-IR also demonstrated its potential as a screening method to identify cinnamon (*C. verum* and *C. cassia*) adulteration in supply chains and to provide accurate and rapid results without sample preparation. In this case, the treatment of spectra using the PLS-DA technique was superior to SIMCA for classification of two types of adulteration: *C. cassia* replaced with spent *C. cassia* and *C. verum* replaced with both *C. cassia* and spent *C. verum* [143]. In the same way, a portable near-infrared spectrometer (NIRS) and an FT-IR benchtop spectrometer were used to detect and quantify the adulterants peanut, pecan, and walnut shells in cumin powder. For both techniques, principal component analysis (PCA) allowed a good class separation (pure cumin vs. adulterated cumin), and later, multivariate analysis SIMCA and PLSR models showed excellent classification and predictive ability. Although FT-IR was superior to the portable NIRS spectrometer, this latter is transportable and cheaper than FT-IR, and could also be implemented along the supply chain as a screening technique [144]. Dhakal et al. [106] used FT-IR spectroscopy to detect Sudan Red and white turmeric (*C. zedoaria*) adulteration in yellow turmeric (*C. longa* L.). A PLSR model for each yellow turmeric—Sudan Red and yellow turmeric—and white turmeric sample was developed to determine the adulterant concentrations, estimating Sudan Red and white turmeric contamination to be in the concentration ranges from 1% to 25% and 10% to 50%, respectively. Recently, Massaro et al. [145] developed and validated a non-targeted method for the authentication of black pepper using NIR coupled to a least absolute shrinkage and selection operator (LASSO), using multiplicative scatter correction (MSC) as a normalization technique.

Raman spectroscopy is a technique based on the inelastic Raman scattering of monochromatic light, which changes when interacting with a sample. This change provides information about vibrational, rotational, and other low-frequency transitions in molecules of the samples. Since Raman is an inherently weak effect, the optical components of a Raman spectrometer must be well-matched and optimized. As source light, monochromatic solid-state laser diodes are often used. This technique can be used to analyse organic samples (solid, liquid, and gas) pre-packaged in plastic or glass, making it feasible for use in bulk industrial applications [146]. In the case of powdered samples like spices, to avoid the interference of intense fluorescence background noise in the spectrum, the technique called surface-enhanced Raman spectroscopy (SERS) is applied. In SERS, the sample is deposited on colloidal or solid metallic surfaces, increasing the signal intensity due to interaction and charge transfer between the adsorbed sample and metallic surface. Chao et al. [147] used Raman imaging and FT-IR spectroscopy to detect Sudan Red and white turmeric adulteration in turmeric powder. The results showed that both IR and Raman spectra can be used to identify Sudan Red contamination in yellow turmeric powder by the development of PLSR models. However, white turmeric Raman peaks overlapped with yellow turmeric peaks, so could not identify this adulteration. Lohumi et al. [118] observed a linear correlation between the resultant peaks in Raman spectroscopy and the concentration of adulterants like Congo red and Sudan I dye. These authors used paprika as a powdered-food model. In a recent study, Zhang et al. [148] developed a precise SERS method to qualify and quantify hydroxy-α-sanshool (α-SOH) in hotpot seasoning, restraining the interference of capsaicin. For this, the samples require pre-treatment with metal–organic frameworks (MOFs), exhibiting an Fe-BTC MOF significant anti-interference effect. 

The hyperspectral imaging (HSI) technique is based on a combination of spectroscopic and imaging techniques. Through the pixel, a hyperspectral image generates a stack of images at each wavelength in the form of a hypercube, providing a large amount of information regarding the physical, textural, and chemical properties of the sample. A spectral hypercube can be constituted by absorbance, transmittance, or reflectance data, among other data types. Because of the type of images, HSI can be applied to monitor the spices during production processes or visualise the selected attributes across the whole sample in a non-destructive manner, giving centralised data about each area for the whole sample [141]. Kiani et al. [100] employed spectral (400–1000 nm) and spatial information of nutmeg extracted using HSI for developing an MLP-ANN method that successfully distinguished non-authenticity in the samples. For cinnamon species’ authentication, NIR-HSI data (1085–1700 nm) were analysed, obtaining a PLS-DA classification model suitable for identifying *C. verum* samples, whereas the model obtained by the support vector machine (SVM) performed better than PLS-DA in identifying *C. cassia* [102]. Wang et al. [149] demonstrated the great potential of HSI technology (VNIR 410–950 nm and SWIR 950–2500 nm) assisted with three regression models, back-propagation neural network (BPNN), partial least squares regression (PLSR), and random forest (RF), to predict the chemical indicators of quality for the spices Red ‘Huajiao’ (*Zanthoxylum bungeanum* Maxim.) and Green ‘Huajiao’ (*Zanthoxylum schinifolium* Sieb. et Zucc.). NIR-HSI has also been presented as a reliable analytical method for the prediction of black pepper adulteration with common adulterant papaya seeds. The PLS model performed with seven important wavelengths of raw spectral data present a good predictive capability for predicting papaya seed concentration (1–30%) in black pepper powder samples [150]. Florián-Huamán [151] used NIR-HSI (900–1710 nm) to detect nut shells in cumin, classifying pure and adulterated samples using SIMCA with an accuracy of 95% for test samples, while the PLSR model was able to successfully predict adulterant concentration. Visible and short wavelengths of near-infrared hyperspectral imaging (Vis-SWNIR-HIS; 400–1000 nm) have been proposed as a novel technique for turmeric authentication and multiple adulterants (corn flour, rice flour, starch, wheat flour, and zedoary) detection [152]. In this study, two multivariate resolution techniques of multivariate curve resolution–alternating least squares (MCR-ALS) and mean-field independent component analysis (MF-ICA) were used to extract pure spatial and spectral profiles of the components.DD-SIMCA and PLS-DA were used for the classification of authentic and adulterated samples.

### 5.5. Other Techniques

#### 5.5.1. Immunology Tests

Immunoassays can be the alternative to expensive chromatographic methods, especially for screening purposes, as they are rapid, sensitive, and specific. The most widely used and effective immunological test for detecting adulterations in many commercial spice products is the enzyme-linked immunosorbent assay (ELISA). Immunological techniques are based on the specific interaction of an antigen with complementary antibodies; this forms the antigen–antibody complex, and helps to identify specific plant material. The result may be, in a qualitative ELISA, a simple positive or negative result for adulteration, or, in a quantitative assay, a measurement of the quantity of the contaminant.

The major limitation of these methods is the affinity of the antibody used, and the development of a highly sensitive immunoassay remains a major challenge [107]. Wang et al. [107] detected rhodamine B, a synthetic colorant that is used illegally to improve the colour of chilli powder, chilli, or sauces, by indirect competitive ELISA, obtaining high sensitivity in the analysis. Also, Oplatowska et al. [108] developed a rapid ELISA method based on highly specific polyclonal antibodies for the detection of Sudan I, a colorant that is fraudulently added to chilli for illegal colour-enhancement purposes. They detected high amounts in three samples and confirmed that the ELISA method allows rapid, sensitive, and high-throughput screening of different food products for the presence of the illegal colorant. Sharma et al. [50] performed a study using two different quantitative sandwich ELISA kits to determine the safety and gluten-free labelling compliance of different food categories including different condiment mixes, chili, curry, and soup mixes. These authors detected levels exceeding the gluten threshold (20 ppm) in some of the condiment samples tested. 

Although several studies have used this technique successfully, some researchers have found that this method produces false positive or false negative results. For example, a false negative for traces of peanut protein in cumin spices has been reported due to the lack of sensitivity of the ELISA kit [48]. For this reason, different authors highlight the need for the use of multiplex techniques and alternative analytical methods to address and/or detect fraud or adulteration in spices, even if elements are present at trace levels.

#### 5.5.2. Microscopy and Sensory Analysis Techniques

Microscopy has proven to be a simple and rapid tool for preliminary screening of the identity and purity of spices by comparing the plant tissues examined with the standard histological characteristics of each spice. Some researchers have used several microscopic techniques, such as light and scanning-electron microscopy, to detect the adulteration of spices and herbs [22].

Zhu and Zhao [109] showed the efficiency of using the microscopic technique to distinguish the micro-morphology characteristics of pure spice powders (cumin, chilli, pepper, and mustard powders) from their adulterants (starch, plant straws, and monosodium glutamate). In addition, the adulteration of black pepper powder with papaya seed powder was identified by meticulous microscopic examination of fatty oils, oil globules, starch granule, fibres, and different features of parenchyma cells [110]. Although microscopic methods can differentiate powdered food products, it has also been argued that it is difficult to analyse powdered samples compared to fresh samples [153]. In some circumstances, it is necessary to use more than one technique to verify results. Ma et al. [111] combined microscopy and GC-MS for the detection of fennel seed adulteration. 

Finally, the sensory analysis technique is also widely used for the detection of fraud or adulteration in spices, as they are well-known for their high organoleptic qualities. Apart from the human panellist, different instrumental techniques have been developed to standardize sensory analysis and detect adulterations in herbs and spices. Techniques such as gas chromatography-olfactometry (GCO) and biosensors such as electronic tongue (E-tongue), electronic nose (E-nose), and electronic eye (E-eye) are various novel techniques used for sensory analysis. These techniques are simple, portable, fast, highly sensitive, and selective. They are based on a set of sensor transducers that can detect complex volatiles present in the headspace of food samples [154].

Kiani et al. [112] used an integrated system to detect saffron adulteration with a computer vision system, an E-nose and a chemometric tool. The results of the analysis revealed that they were able to effectively differentiate authentic saffron samples from adulterated ones. based on their aroma intensity. The ability of E-nose and Voltammetric electronic tongue (VE-tongue) in combination with Solid Phase Microextraction-Gas chromatography–Mass Spectrometry (SPME-GC-MS), along with chemometrics to discriminate cumin samples from adulterated ones and those with different geographical origins was also demonstrated [113]. The results indicated that VE-tongue has more potential (100% accuracy) for detection and discrimination than the other methods. Nanotechnology-based E-nose was used to detect safflower and corn-stigma adulteration in saffron and other spices [114]. The results revealed that the system can successfully recognize saffron adulteration with 100% accuracy and is able to successfully differentiate unadulterated saffron from adulterated saffron which has an adulteration level of more than 10%.

## 6. Adulteration Prevention Measurement for Spices and Herbs 

Standardised control systems in the food chain have in recent decades focused on the mitigation of physical and chemical, and especially biological hazards. This lack of systematisation for fraud control is one of the causes of the generation of vulnerabilities within the food marketing chain, generating opportunities for illicit practices such as fraud and adulteration. The spice industry particularly is one of the most affected by these criminal practices [155]. Guarantees of the authentication of food products represent a growing demand among consumers, who express the importance of knowing the origin of the products and the processes undergone [156], in addition to considering access to technological tools for self-authentication as appropriate [157]. As a result, systematic fraud control measures have emerged in recent years as fundamental aspects of quality assurance for agri-food products. Based on risk analysis, different food safety standards have been developed in recent years (reviewed by [158,159]). Among these, the Campden Threat Assessment and Critical Control Point (TACCP) and Vulnerability Assessment and Critical Control Point (VACCP), defined by PAS96:2017, can be highlighted; they are oriented to the identification of threats and malicious activities on the one hand, and to the identification of vulnerabilities on the other hand. In the same way, with methodologies based on risk analysis, guides have been published for the detection and management of vulnerabilities associated with food fraud, such as *The Global Standard for Food Safety*, *version 9*, by the British Retail Consortium (BRC 2022) and *The IFS Standards Product Fraud—Guidelines for Implementation*. In addition, there are online tools for carrying out an initial screening on the vulnerabilities of an industry or supply chain, by reviewing previous incidents or suspected fraud through the Food Fraud Initial Screening Model (FFISM) [160] or through a prior analysis of the ingredients, products, or product groups in terms of motivation, opportunity, and control for fraudulent activities offered by the food fraud vulnerability self-assessment tool (SSAFE FFVA tool; [161]). In the specific case of the spice sector, Silvis et al. [162] analysed the vulnerability of this sector by applying the SSAFE FFVA tool, and interviewing eight companies involved in the spice marketing chain. In terms of the greatest vulnerabilities detected, the responses in the opportunities dimension indicated the ease of adulteration and the difficulty of detecting adulteration, especially in ground spices and herbs. Similarly, difficulties were detected in counterfeiting. These aspects were reflected in the perception of low transparency in the early stages of the chain. An effective measure to reduce opportunities for fraud and adulteration is to purchase whole spices. Regarding motivational aspects, high competition in the sector, together with the high overall unit price of spices and valuable components or attributes of raw materials generate high vulnerability. Finally, the levels of corruption in the countries and suppliers with which companies operate are detected as high vulnerabilities. Considering that illicit practices can occur throughout the entire supply chain, standards to mitigate fraud and adulteration must be adopted across the board by all economic operators, regardless of the size of the operators. In this respect, it has been found that specific fraud control measures are not common in small companies. In general, vulnerabilities in controls were associated with suppliers, despite the fact that the companies themselves did not have in-house-tampering control systems in place. In terms of soft controls, the legislative framework, and the limited attention of official control to fraudulent practices do not help to reduce the vulnerabilities of the sector.

## 7. Future Perspectives and Conclusions

There is a growing fraud and adulteration incidence in spices and herbs. This has been highlighted by the increasing number of RASFF alerts and the number of scientific articles published based on the Web of Science databases during the last decades. The most frequent and important type of adulteration refers to the use of different unauthorized synthetic dyes; the most used are the Sudan dyes in spices that stand out for their colour and colour intensity (curry, chili pepper, paprika, among others). According to data from the European Commission regarding the use of inappropriate handling and fraudulent practices, the most adulterated herb was oregano, compared with other plant species. The economic relevance around the market of these ingredients is quite profitable, since the spices and herbs are valued ingredients, desirable for consumers due to their sensorial and functional activities. In recent years, research has been carried out to develop methods able to distinguish between authentic and adulterated spices and herbs, with DNA-based techniques and mainly spectroscopy and image analysis methods being the most-recommended ones. Although sensitive and reliable methods have been optimised in this sense, future efforts should be made in order to improve their rapidness, avoiding the destruction of samples and even ensuring that they can be implemented in production lines, allowing control over all or most of the processed products in order to ensure the authenticity of herbs and spices and the establishment of effective adulteration prevention measures. Finally, the implementation of on-site and/or consumer-friendly detection technologies would be desirable, considering the willingness of educated consumers who would like to use a device to validate food label contents.

## Figures and Tables

**Figure 1 foods-12-03373-f001:**
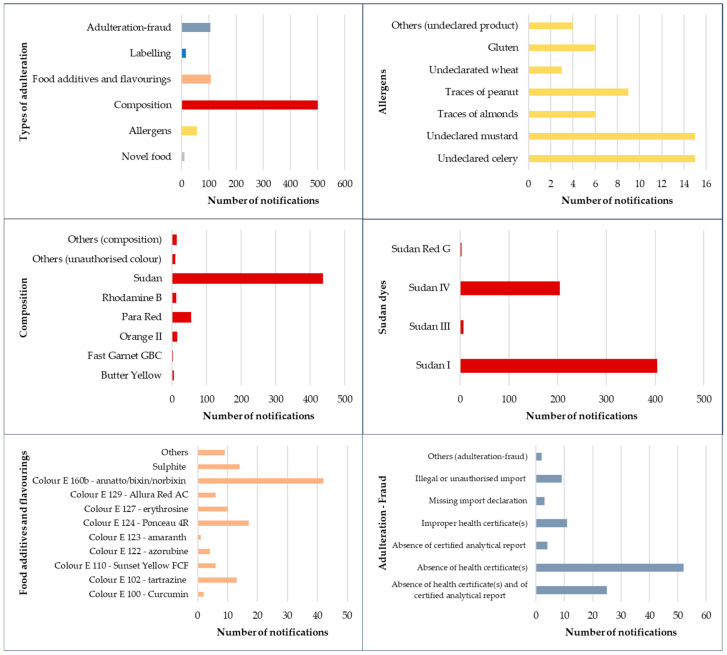
Notifications about potential risks encountered in herbs and spices during a period of 33 years (1989–2020). Types of notifications: in blue, fraud-adulterations; in red, addition of unauthorised/declared compounds; in pink, addition of unauthorised/declared additives; in yellow, presence of allergens; in grey, presence of unauthorised novel foods. Source: [58].

**Figure 2 foods-12-03373-f002:**
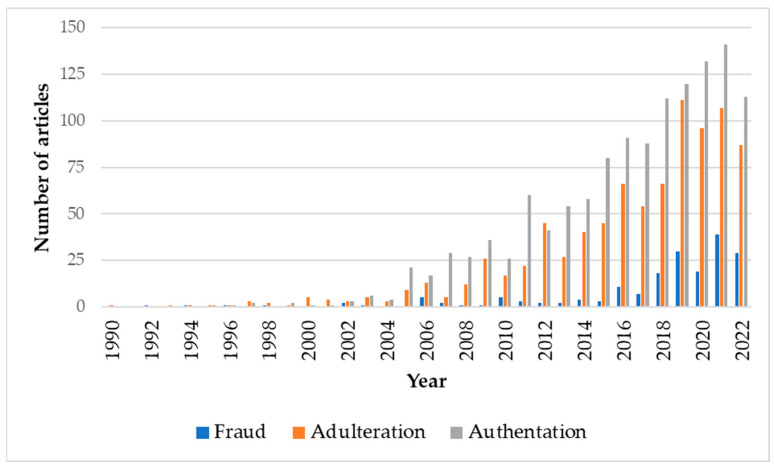
Articles on herbs and spices related to fraudulent practices in 32 years (1990–2022). Source: [60].

**Table 1 foods-12-03373-t001:** World area harvested, production and imports of the main herbs and spices in 2021.

Herbs and Spices	Area Mill. ha	Production Mill. t	Imports USD Mill.	Main Producers
Anise, badian, coriander, cumin, caraway, fennel, and juniper berries	2.30	2.70	1398.58	India, Türkiye, Mexico, Russia, Syria
Chillies and peppers, dry (*Capsicum* spp.)	1.62	4.84	2773.00	India, Bangladesh, Thailand, China, Ethiopia
Cinnamon and cinnamon-tree flowers	0.30	0.23	1048.17	China, Indonesia, Vietnam, Sri Lanka, Madagascar
Cloves (whole stems)	0.67	0.19	451.52	Indonesia, Madagascar, Tanzania, Comoros, Sri Lanka
Ginger	0.45	4.90	1506.24	India, Nigeria, China, Indonesia, Nepal
Mustard seed	0.63	0.53	316.37	Nepal, Russia, Canada, Myanmar, Ukraine
Nutmeg, mace, cardamoms	0.47	0.15	1452.57	India, Indonesia, Guatemala, Nepal, Sri Lanka
Other stimulant, spicy and aromatic crops	1.41	3.15	2236.67	India, Ethiopia, Türkiye, Bangladesh, Yemen
Pepper (*Piper* spp.)	0.68	0.79	2025.28	Vietnam, Brazil, Indonesia, Burkina Faso, India
Peppermint, spearmint	0.00	0.04	3.67	Morocco, Argentina, Mexico, Japan, Georgia
Sesame seed	12.51	6.35	3570.79	Sudan, India, Tanzania, Myanmar, China
Vanilla	0.09	0.01	901.44	Madagascar, Indonesia, Mexico, Papua New Guinea, China

Source: [16,17] (https://www.fao.org/statistics/en/, accessed on 23 August 2023).

**Table 2 foods-12-03373-t002:** World production and economic data of the main herbs and spices in 2021.

Continent	Production (Mill. t)	Exports (Mill. t)	Imports (Mill. t)	Exports (USD Mill.)	Imports (USD Mill.)
Africa	6.27	1.40	0.35	2631.21	733.56
America	0.90	0.72	0.89	2028.27	3085.37
Asia	16.39	4.09	4.66	9720.66	9316.54
Europe	0.30	0.76	1.28	2563.04	4369.02
Oceania	0.02	0.01	0.04	69.47	179.81
**World**	**23.87**	**6.97**	**7.22**	**17,012.65**	**17,684.30**

Source: [17] (https://www.fao.org/statistics/en/, accessed on 23 August 2023).

**Table 3 foods-12-03373-t003:** Protected Designation of Origin (PDO) and Protected Geographical Indication (PGI) in European Union and non-EU countries.

Type and Country Zone	Product Type	Name	Country
Protected Designation of Origin (PDO) European Union	Saffron	Krokos Kozanis	Greece
		Azafrán de la Mancha	Spain
		Zafferano dell’Aquila; Zafferano di San Gimignano; Zafferano di Sardegna	Italy
	Paprika	Pimentón de La Vera; Pimentón de Murcia; Pimentón de Mallorca	Spain
		Piment d’Espelette	France
		Szegedi paprika; Kalocsai fűszerpaprika-őrlemény	Hungary
		Žitavská paprika	Slovakia
	Cumin	Český kmín	Czechia
Protected Geographical Indication (PGI) European Union	Thyme	Thym de Provence	France
	Vanilla	Vanille de l’île de La Réunion	France
Protected Geographical Indication (PGI) non-EU countries	Pepper	Poivre de Penja	Cameroon
	Pimienta	Poivre de Kampot	Cambodia
	Ginger	Luoping Xiao Huang Jiang	China
	Cinnamon	Ceylon Cinnamon	Sri Lanka

Source: [19] (https://ec.europa.eu/agriculture/eambrosia/geographical-indications-register/, accessed on 23 August 2023).

**Table 4 foods-12-03373-t004:** Notifications about potential risks encountered in herbs and spices (2020–2022). Source: [58].

**Spices and Herbs**	**Product**	**Adulteration (Novel Food Ingredient)**	**Date**	**Notifying Country**	**Distribution**	**Origin**
Oregano	Oregano	Novel food olive leaf	2020	Germany	Netherlands, France, Austria, Switzerland, Algeria, Germany	Türkiye
Spices	Spice mix	*Angelica sinensis*	2020	Finland	Hong Kong, Finland, Netherlands	Hong Kong
**Spices and Herbs**	**Product**	**Adulteration (Undeclared Product or Other Botanical Plants or Allergens)**	**Date**	**Notifying Country**	**Distribution**	**Origin**
Basil	Basil	Celery	2022	Cyprus	Cyprus	Greece
Coriander	Coriander	Mustard	2021	Netherlands	Netherlands	Ukraine
	Coriander	Mustard	2022	Spain	Portugal, Sweden, Belgium, Bulgaria, Andorra, Spain, Chile, France, Colombia, Panama, Costa Rica, Italy, Honduras, Lithuania, Mexico	Spain
Cumin	Cumin (ground)	Gluten	2020	European Commission	Spain, Andorra	Spain
	Cumin (ground)	Mustard	2021	Spain	Spain	France
	Cumin (ground)	Sesame	2022	Spain	Dominican Republic, Portugal, Bulgaria, Switzerland, United States, Germany, Andorra, Spain, France, United Kingdom, Guinea, Mexico	Spain, India
Curry	Curry (Madras curry)	Gluten	2021	Spain	Spain, Portugal	Spain
	Curry (powder)	Traces of peanut		European Commission	Northern Ireland	India
Spearmint	Spearmint (crushed)	Celery	2022	Cyprus	Cyprus	Egypt
Spices	Spice mix	Mustard	2021	Spain	Spain, France, Portugal	Spain
	Spice mix	Traces of gluten, mustard and lupin		Netherlands	Netherlands, Aruba, Lithuania	Netherlands
**Spices and Herbs**	**Product**	**Adulteration (Composition: Illegal Dyes)**	**Date**	**Notifying Country**	**Distribution**	**Origin**
Cumin	Cumin	Auramine O and cis-Bixin	2022	Lithuania	Spain, Portugal, Greece, Czech Republic, Lithuania	India
Curry	Curry	Sudan I	2021	Austria	Austria, Hungary	Türkiye
	Curry	Rhodamine B		Austria	Netherlands, France, Austria, Germany	Türkiye
	Curry	Sudan I		Netherlands	Netherlands, Belgium	India
	Curry (powder)	Sudan I and Sudan IV		Belgium	Distribution restricted to notifying country	Türkiye
	Curry (powder)	Orange II	2022	Belgium	Belgium	Cameroon
Pepper	Chilli pepper (powder)	Orange II	2020	Belgium	Belgium	Cameroon
	Pepper (dried)	Orange II		Belgium	Germany	Cameroon
	Pepper (grind dried)	Orange II		Belgium	United Kingdom	Nigeria
	Cayenne pepper (powder)	Orange II and Sudan I		Belgium	United Kingdom	Nigeria
	Chilli pepper (powder)	Orange II	2021	Belgium	Germany	Ghana
	Chilli pepper (powder)	Orange II		Belgium	Netherlands, Belgium	Nigeria
	Chilli pepper (powder)	Orange II		Belgium	Germany	Togo
	Chilli pepper (powder)	Orange II and Sudan I		Belgium	France	Togo
	Chilli pepper (powder)	Sudan I, Sudan IV and Rhodamine B	2022	Belgium	Product not (yet) placed on the market	Bangladesh
	Chilli crushed with seed	Sudan I, Sudan III and Sudan IV		Germany	Germany, Italy, Luxembourg	Unknown
	Pepper (crushed)	Sudan I and Sudan IV		Switzerland	Distribution restricted to notifying country	China
Sumac	Sumac (ground)	Sudan I and Sudan Orange G	2022	Latvia	Latvia, Russia, Poland, Croatia	Türkiye
	Sumac (ground)	Sudan IV		Switzerland	Switzerland	Türkiye
	Sumac (ground)	Sudan IV		Switzerland	Distribution restricted to notifying country	Türkiye
	Sumac	Sudan IV		Germany	Denmark, Poland, France, Sweden, Slovenia, Austria, Germany	Türkiye
Spices and herbs	Spices	Sudan I	2020	Latvia	Latvia	Uzbekistan
	Spices and herbs	Sudan I	2021	Latvia	Uzbekistan	Russia
	Spices	Sudan IV	2022	Belgium	France, Belgium, Germany	France
	Spices (couscous spice mix)	Sudan I and Sudan IV		Switzerland	Latvia, Malta, Netherlands, Portugal, Romania, Austria, Belgium, Switzerland, Germany, Denmark, Spain, France, Italy, Lithuania	Lebanon
	Spices (spice preparation)	Sudan IV		Switzerland	Switzerland	Türkiye
	Spices	Sudan II, Sudan III and Sudan IV		Latvia	Latvia	Russia
	Herbs (Italian product “granelli d’erbe”)	Aloe-emodin and emodin		France	France	Italy
Turmeric	Turmeric	Residue of lead and Sudan I	2022	Belgium	Product not (yet) placed on the market	Bangladesh
**Spices and Herbs**	**Product**	**Adulteration (Undeclared or Unauthorised Food Additives and Flavourings)**	**Date**	**Notifying Country**	**Distribution**	**Origin**
Cinnamon	Cinnamon	Sulphite	2021	Belgium	Northern Ireland, Austria, Belgium, Switzerland, Czech Republic, Germany, Denmark, Spain, Finland, France, Greece, Croatia, Hungary, Ireland, Iceland, Italy, Lithuania, Luxembourg, Latvia, Netherlands, Norway, Poland, Portugal, Romania, Sweden, Slovenia, Slovakia	United Kingdom
	Cinnamon	Sulphite		Belgium	Denmark, Netherlands, France, Sweden, Belgium, Italy, Germany	Sri Lanka
Cumin	Cumin (powder)	Colour E104—Quinoline yellow	2022	Spain	Spain	India
	Cumin	Colour E 160b—annatto/bixin/norbixin	2022	Lithuania	Spain, Portugal, Greece, Czech Republic, Lithuania	India
Herbs	Dried lily bulbs	E220—Sulphur dioxide	2021	Denmark	Denmark, Spain, Netherlands, France, Austria, Hungary, Germany	China
Spices	Tandoori masala	Colour E 102—tartrazine and colour E 129—Allura Red	2022	Denmark	Denmark	Spain
Sumac	Spice preparation	Colour E 122—azorubine and Colour E 124—Ponceau 4R	2022	Switzerland	Switzerland	Türkiye
**Spices and Hrbs**	**Product**	**Adulteration (Labelling Absent)**	**Date**	**Notifying Country**	**Distribution**	**Origin**
Ginger	Ginger (powder)	Missing allergen labelling	2022	Germany	Denmark, Latvia, Estonia, Finland, Poland, Germany	India

## Data Availability

Data is contained within the article.

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
