# Peer review of "Spice and Herb Frauds: Types, Incidence, and Detection: The State of the Art"

_foods, 2023, doi:10.3390/foods12183373_

Round 1

Reviewer 1 Report

Manuscript title : Detection of the spice and herb frauds: State of the art. It is not the state of the art. Authors only discussed detection technology. Authors should include all suggestions below and revise accordingly. Many relevant studies are missing

Abstract

Authors should provide enough background on the review and need of performing the review

Review findings should be included in abstract

Conclusions should be included in abstract

Citation is not appropriate format

1.3. Type of frauds and 1.3. Fraud incidence. Is it 1.4?

Figure 1: quality must be improved. As such, the text is not readable

Table 4. Alerts of herbs and spices (2020-2022).. what do authors mean by alerts? Please revise it

There are many sections on 1.3. please revise it accordingly

Methods for detecting spice and herbs frauds: This section must be improved by providing a table containing an overview of all methods and how to use different detection methods, advantages and disadvantages.

Provide an example of each technique (in figure) by taking any spice/herb. This way provides a best understanding of the techniques. Author must include figures

Also, another section must be introduced by discussing what are the sources that can be used to adulterate spices/herbs with Table Like cinnamon is adulterated with fake cinnamon. Like this turmeric, etc Authors should discuss this and provide what are the difference technologies used to prevent these or identify

Authors must introduce section dealing with adulteration prevention measurement for spices/herbs

Also, authors must discuss on-site and/or consumer-friendly detection technologies

Conclusions should be revised after including the above suggestions

References must be in line with J format 

Author Response

Manuscript ID: foods-2562659

Type: Review

Title: Detection of the spice and herb frauds: State of the art

Reviewer 1

Manuscript title: Detection of the spice and herb frauds: State of the art. It is not the state of the art. Authors only discussed detection technology. Authors should include all suggestions below and revise accordingly. Many relevant studies are missing.

Response: We have included the suggestions made by Reviewer 1 in the revised version of the manuscript.

Abstract

Authors should provide enough background on the review and need of performing the review. Review findings should be included in abstract.

Conclusions should be included in abstract.

Response: Abstract and Conclusions have been revised as requested by Reviewer 1.

Citation is not appropriate format.

Response: According to instructions for authors of the journal (https://www.mdpi.com/journal/foods/instructions). Foods now accepts free format submission, and it is literally written in the instructions:

“Your references may be in any style, provided that you use the consistent formatting throughout. It is essential to include author(s) name(s), journal or book title, article or chapter title (where required), year of publication, volume and issue (where appropriate) and pagination. DOI numbers (Digital Object Identifier) are not mandatory but highly encouraged. The bibliography software package EndNote, Zotero, Mendeley, Reference Manager are recommended”. Therefore, authors followed the instructions of authors of the journal to prepare the manuscript.

1.3. Type of frauds and 1.3. Fraud incidence. Is it 1.4?

Response: Yes, it is. This has been amended in the revised version of the manuscript.

Figure 1: quality must be improved. As such, the text is not readable

Response: As suggested by Reviewer 1, the quality of Figure 1 has been improved.

Table 4. Alerts of herbs and spices (2020-2022).. what do authors mean by alerts? Please revise it

Response: Sorry, Reviewer 1 is right. The Table header has been corrected in the revised manuscript. We have replaced the word “alert” with “notification”.

There are many sections on 1.3. please revise it accordingly

Response: We do not understand this from Reviewer 1 since there are no sections on 1.3. We included the different types of fraud and the fraudulent adulteration practices in different bullet points but that’s all. We consider that it is necessary for readers’ understanding of the different types of fraud and why it is necessary to detect such frauds.

Methods for detecting spice and herbs frauds: This section must be improved by providing a table containing an overview of all methods and how to use different detection methods, advantages and disadvantages.

Response: As requested by Reviewer 1, a Table (Table 5) containing an overview of all methods including the main results has been included in the revised manuscript. The advantages and disadvantages of the methods have been provided in the corresponding sections.

Provide an example of each technique (in figure) by taking any spice/herb. This way provides a best understanding of the techniques. Author must include figures.

Response: As suggested by Reviewer 1, we have provided a new Figure (Figure 3) that overviews the different techniques that can be employed to detect adulteration or fraud in paprika or chili powder, two of the spices most frequently adulterated according to the RASFF database.

Also, another section must be introduced by discussing what are the sources that can be used to adulterate spices/herbs with Table Like cinnamon is adulterated with fake cinnamon. Like this turmeric, etc Authors should discuss this and provide what are the difference technologies used to prevent these or identify.

Response: We do not consider that we should introduce another section in the revised version of the manuscript since the information asked for Reviewer 1 is displayed in previous sections 1.4 and 1.5 (now sections 2 and 3).

Specifically, in section 2 (types of frauds) we have included numerous examples about fraudulent adulteration practices (ESA, 2018b), some of them are shown below:

  • A different part of the same botanical plant, rather than the one declared, to the extent that this would mislead the customer (line 167): … “for example, the addition of non-spice plant matter such as stamens and safflower in pure saffron (Petrakis & Polissiou, 2017; Soffritti et al., 2016)” (lines 171-174).
  • Technically avoidable amounts of parts from other botanical plants than the one declared (line 178): … “There are some examples, in the case of black pepper (Piper nigrum L.), one of the most widely used spices in the world, which can be adulterated with papaya seeds (Carica papaya L.), that have very similar external characteristics (Hoffman et al., 2021)” (lines 182-184).
  1. Ingredients, additives, dyes, or any other constituent not approved for use in herbs and spices (line 189): … “For example, the use of cereal and potato starch is included in some spice powders such as paprika, curry, turmeric, and ginger. In addition, corn starch is added as an adulterant in onion powder (Lee et al., 2015; Lohumi et al., 2014), garlic and ginger powders (Lee et al., 2014)” (lines 191-194).
  2. Ingredients, additives, dyes, or any other constituent approved for use in food but unlawfully not declared or indicated in a form which might mislead the customer (lines 216-217): … “Ground chilli spice can be adulterated with dried red beet pulp and powdered Ziziphus nummularia (Burm.f.). fruits (Dhanya et al., 2011) or with groundnut or almond shell residues which can cause health related issues to the consumer” (lines 224-226).
  3. Herbs and spices that have had any valuable constituent omitted or removed which misleads the customer (e.g., spent and partially spent herbs and spices, de-oiled material, defatted material) (lines 233-234): … “Adulteration is notable with the addition of spices that have had their valuable components removed, such as the inclusion of defatted paprika to paprika (ASTA, 2011). Paprika oil (oleoresin) is a quality product with multiple health benefits. However, once removed from paprika, the remaining spent product can be used to adulterate paprika. Once this oleoresin is removed from paprika, the remaining "spent" material is a waste product (Galvin-King et al., 2020)” (lines 237-240).

In section 3 authors have analysed the health notifications provided by the RASFF system for the historical series from 1989 to 2020 (RASFF, 2023) in relation to herbs and spices, showing the different types of adulterations and frauds, analysing the adulterants in detail (lines 256-312). In addition, authors showed a Figure (Figure 1) where an overview of this information can be visualized. Furthermore, authors have analysed information obtained by health notifications encountered in herbs and spices provided by RASFF in the period 2020-2022 (lines 316-327; Table 4). This information about spices and herbs and their adulterants is very profoundly detailed in Table 4. We consider that the information provided in this revision is very new and updated and readers can understand perfectly which ones are herbs and spices that can be adulterated and how.

Authors must introduce section dealing with adulteration prevention measurement for spices/herbs

Response: As requested by Reviewer 1, we have introduced a section dealing with adulteration prevention measurement for spices and herbs in the revised manuscript (section 6: Adulteration prevention measurement for spices and herbs).

Also, authors must discuss on-site and/or consumer-friendly detection technologies

Response: We had included some of on-site detection technologies in the previous version of the manuscript. As Reviewer 1 suggested, we have included some of this in the new section 6, regarding consumer-friendly detection technologies, and in section 7 (Future perspectives and conclusions), highlighting the necessity to increase efforts to implement and improve on-site and/or consumer-friendly detection technologies.

Conclusions should be revised after including the above suggestions

Response: Conclusions have been revised considering the changes included in the revised manuscript.

References must be in line with J format 

Response: According to instructions for authors of the journal (https://www.mdpi.com/journal/foods/instructions). Foods now accepts free format submission, and it is literally written in the instructions:

“Your references may be in any style, provided that you use the consistent formatting throughout. It is essential to include author(s) name(s), journal or book title, article or chapter title (where required), year of publication, volume and issue (where appropriate) and pagination. DOI numbers (Digital Object Identifier) are not mandatory but highly encouraged. The bibliography software package EndNote, Zotero, Mendeley, Reference Manager are recommended”. Therefore, authors followed the instructions of authors of the journal to prepare the manuscript.

Reviewer 2 Report

There is a growing fraud and adulteration incidence in spices and herbs. This has 857 been highlighted by the increasing number of RASFF alerts during the last decades. The 858 economic relevance around the market of these ingredients is quite profitable since the 859 spices and herbs are valued ingredients desirable for consumers due to their sensorial and 860 functional activities.

The work done by the author's team looks very large-scale and important.

The introduction is logically structured, fully reflects the objectives of the study.

Methodological aspects are described in full.

The tables include the necessary data.

In my opinion, a very interesting study and a high-quality manuscript but its plagiarism will be reduced now its 30%

Minor editing of English language required

Author Response

Manuscript ID: foods-2562659

Type: Review

Title: Detection of the spice and herb frauds: State of the art

Reviewer 2

There is a growing fraud and adulteration incidence in spices and herbs. This has been highlighted by the increasing number of RASFF alerts during the last decades. The economic relevance around the market of these ingredients is quite profitable since the spices and herbs are valued ingredients desirable for consumers due to their sensorial and functional activities.

The work done by the author's team looks very large-scale and important.

The introduction is logically structured, fully reflects the objectives of the study.

Methodological aspects are described in full.

The tables include the necessary data.

Response: We thank Reviewer 2 for the positive comments on the manuscript.

In my opinion, a very interesting study and a high-quality manuscript but its plagiarism will be reduced now its 30%

Response: Very respectfully, we are very surprised by this. We have used the anti-plagiarism program of our University and we have obtained less than 10% plagiarism (avoiding references). The only part of the text that is literally copied is the ones included in section 2 related to current legislation regarding types of fraud (Regulation EU 2017/625) and fraudulent practices described by European Association Spices (ESA, 2018b). We believe that it is very important that the reader knows the types of fraud and adulteration and the fraudulent practices that can be carried out in herbs and spices and why they can be carried out. Anyway, if you still consider that some parts should be rewritten, please let us know and we will work on these parts of the text.

Reviewer 3 Report

The review entitled “Detection of the spice and herb frauds: State of art”, written by Velázquez et al aims at summarizing the various types of herbs and spices adulterations and frauds and the different analytical methods used to detect them.  

This manuscript falls in the scope of Foods, it is very well written, organized, and comprehensively described, and represents an essential contribution to the subject. However, some minor revisions and corrections are needed before being accepted for publication.

-          Define RASFF (do not use abbreviations in the abstract)

-          Line 24 – correct “imagen” – “image analysis”

-          The introduction section is too long comprising several subsections. I suggest renumbering. For example, the introduction should include subsections 1.1 (Herbs and spices sensorial and healthy-related properties), 1.2 (World market of herbs and spices: economic relevance), and 1.3 (Quality protection of herbs and spices).  The section describing the types of fraud should be section 2, etc. Moreover, there is an error in the numbering of these subsections (all of them have the same number).

-          Since an essential part of the review is focused on the various types of frauds and adulterations of herbs and spices I suggest improving the Review title in this regard.

-          Lines 29 – 31 – This sentence should be rewritten.

-          Lines 31 -32 – “There is a great diversity of plant species” – do you mean “There is a great diversity of herbs and spices?

-          A paragraph reporting the use of herbs and spices in several traditional medical systems is missing (eg. ayurvedic medicine). Please add.

-          Line 64 – FAO - Food and Agriculture Organization of the United Nations

-          Lines 71 – 73 – This sentence is not necessary – Table 1 is already introduced and its title is clear. Please delete.

-          Line 356 – “saffron “ is repeated

-          Table 4 – Angelica sinensis in italics

-          Lines 434 – 436, 479,  and 490 – 491 – general information – please remove

-          Lines 480 – 481 – this sentence is not understandable. Please revise.

-          Lines 484 – 488 – the paper is not about adulteration of oils and fats. Please remove the sentences.

-          Line 524 – Vitex negundo in italics

- Section 3 - Future perspectives and conclusions - should be improved by making a critical assessment of all the collected data, highlighting the main advantages and disadvantages of the different analytical techniques for the detection of herbs and spices fraud. This improved discussion will help the reader to make decisions about further studies directions, and what is already shown to be promising and not promising.

Author Response

Manuscript ID: foods-2562659

Type: Review

Title: Detection of the spice and herb frauds: State of the art

Reviewer 3

The review entitled “Detection of the spice and herb frauds: State of art”, written by Velázquez et al aims at summarizing the various types of herbs and spices adulterations and frauds and the different analytical methods used to detect them. 

This manuscript falls in the scope of Foods, it is very well written, organized, and comprehensively described, and represents an essential contribution to the subject. However, some minor revisions and corrections are needed before being accepted for publication.

Response: We thank Reviewer 3 for the positive comments on the manuscript.

Define RASFF (do not use abbreviations in the abstract)

Line 24 – correct “imagen” – “image analysis”

Response: This has been corrected in the revised version of manuscript.

The introduction section is too long comprising several subsections. I suggest renumbering. For example, the introduction should include subsections 1.1 (Herbs and spices sensorial and healthy-related properties), 1.2 (World market of herbs and spices: economic relevance), and 1.3 (Quality protection of herbs and spices).  The section describing the types of fraud should be section 2, etc. Moreover, there is an error in the numbering of these subsections (all of them have the same number).

Response: We have renumbered the sections of the manuscript as suggested by Reviewer 3.

Since an essential part of the review is focused on the various types of frauds and adulterations of herbs and spices I suggest improving the Review title in this regard.

Response: The title has been revised according to Reviewer 3 suggestions. The new title of the manuscript is “Spice and herb frauds: types, incidence, and detection. State of the art.”

Lines 29 – 31 – This sentence should be rewritten.

Response: This sentence has been rewritten.

Lines 31 -32 – “There is a great diversity of plant species” – do you mean “There is a great diversity of herbs and spices?

Response: This has been corrected in the revised manuscript.

A paragraph reporting the use of herbs and spices in several traditional medical systems is missing (eg. ayurvedic medicine). Please add.

Response: A paragraph mentioning that herbs and spices are used in several traditional medical systems has been added in section 1.1. of the revised manuscript.

Line 64 – FAO - Food and Agriculture Organization of the United Nations

Lines 71 – 73 – This sentence is not necessary – Table 1 is already introduced and its title is clear. Please delete.

Line 356 – “saffron “ is repeated

Table 4 – Angelica sinensis in italics

Lines 434 – 436, 479,  and 490 – 491 – general information – please remove

Lines 480 – 481 – this sentence is not understandable. Please revise.

Lines 484 – 488 – the paper is not about adulteration of oils and fats. Please remove the sentences.

Line 524 – Vitex negundo in italics

Response: The abovementioned suggestions have been corrected.

Section 3 - Future perspectives and conclusions - should be improved by making a critical assessment of all the collected data, highlighting the main advantages and disadvantages of the different analytical techniques for the detection of herbs and spices fraud. This improved discussion will help the reader to make decisions about further studies directions, and what is already shown to be promising and not promising.

Response: The section entitled Future perspectives and conclusions has been revised and improved taking into account the comments of Reviewers. We did not focus this section on the advantages and disadvantages of the different techniques, but more on the most frequent types of frauds and adulterants in spices and herbs lately, and what’s next in the future in this field research including the use of on-site and consumer-friendly detection technologies.

Author Response

Manuscript ID: foods-2562659

Type: Review

Title: Detection of the spice and herb frauds: State of the art

Reviewer 4

This paper describes frauds that can occur in the herb and spice trade and the analytical methods available for their detection. The paper is interesting and well written and presented and free of typographical errors.

Response: We thank Reviewer 4 for the positive comments on the manuscript.

Specific comments:

  1. For FTIR analysis, are the samples analysed as the solid or as the extract?

Response: For FTIR analysis, the samples are solid with only sifted treatment. This has been included in the revised manuscript (line 589).

  1. Authors can comment on the extent that a high quality herb (presumably with higher concentrations of putative actives) is diluted with the same herb of lower quality, keeping in mind that natural variations occur (due to growing conditions, time of harvest, post harvest and so on). How would this type of adulteration be handled?

Response: This type of adulteration is one of the most difficult to detect. It is true that geographical and growing conditions, varieties, time of harvest, and post-harvest can affect the quantity and even the profile of bioactive compounds of a determined herb; however, this adulteration can be handled if a minimum concentration (a threshold) of the putative actives of a specific herb is known considering the mentioned factors. Adulterated herbs would have less quantity of putative actives or even a different bioactive profile than the high-quality herb and they will be discarded. This has been included in section 2 of the revised manuscript.

  1. I would have liked to see a mention of statistical methods like principal component analysis and scatter plots to help in identifying herb quality.

Response: As requested by Reviewer 3, a mention of some explorative statistical techniques has been included in the revised version of the manuscript (section 5.4., lines 563-565; 597-598; 605-607). This type of technique is used to obtain an overview of the data set, its variance, possible outliers, or influential variables in addition to a visualization of the data.

Round 2

Reviewer 1 Report

The quality of the manuscript is now improved. In my opinion, this version can be accepted for possible publication consideration in Foods.

The quality of the manuscript is now improved. In my opinion, this version can be accepted for possible publication consideration in Foods.